# Extant and extinct bilby genomes combined with Indigenous knowledge improve conservation of a unique Australian marsupial

**A list of authors and their affiliations appears at the end of the paper**

Ninu (greater bilby, *Macrotis lagotis*) are desert-dwelling, culturally and ecologically important marsupials. In collaboration with Indigenous rangers and conservation managers, we generated the Ninu chromosome-level genome assembly (3.66 Gbp) and genome sequences for the extinct Yallara (lesser bilby, *Macrotis leucura*). We developed and tested a scat single-nucleotide polymorphism panel to inform current and future conservation actions, undertake ecological assessments and improve our understanding of Ninu genetic diversity in managed and wild populations. We also assessed the beneficial impact of translocations in the metapopulation ($N$ = 363 Ninu). Resequenced genomes (temperate Ninu, 6; semi-arid Ninu, 6; and Yallara, 4) revealed two major population crashes during global cooling events for both species and differences in Ninu genes involved in anatomical and metabolic pathways. Despite their 45-year captive history, Ninu have fewer long runs of homozygosity than other larger mammals, which may be attributable to their boom–bust life history. Here we investigated the unique Ninu biology using 12 tissue transcriptomes revealing expression of all 115 conserved eutherian chorioallantoic placentation genes in the uterus, an $XY_1Y_2$ sex chromosome system and olfactory receptor gene expansions. Together, we demonstrate the holistic value of genomics in improving key conservation actions, understanding unique biological traits and developing tools for Indigenous rangers to monitor remote wild populations.

Bilbies are unique marsupials and are the only members of the family Thylacomyidae. They include the extant greater bilby (*Macrotis lagotis*; Fig. 1a) and the extinct lesser bilby (*Macrotis leucura*; Fig. 1b). Bilbies are culturally important to Indigenous Australians, with their common name derived from the Yuwaalaraay word, Bilba. The many First Nations across Australia have different names for bilby (Extended Data Table 1), but here we use Ninu to represent the greater bilby as this is the name used by the Kiwirrkurra community (where most of

our wild samples are from) and Yallara for the lesser bilby. Bilbies were once an important meat source for desert people, and their valuable long black tails with white fluffy ends were used in cultural practices associated with their deep symbolism in love and marriage[1]. Indigenous knowledge, bilby songlines, ceremonies and stories exist across Australia, linking sites and people. Their strong connection to the species continues even in areas where bilbies are now locally extinct. Loss of this Indigenous knowledge and land management practices

✉ e-mail: carolyn.hogg@sydney.edu.au

**Fig. 1 | Historical and contemporary distributions of both the Ninu and Yallara and phylogenetic relationships between the two species. a**, Photo of a Ninu with its large ears and characteristic tail; photo credit: E. Peel. A map showing the historical Ninu range (light blue) and estimated current range (dark blue); temperate and semi-arid sampling locations are noted. **b**, Photo of mounted Yallara specimen; photo credit: K. Travouillon. A map showing the historical Yallara range. **c**, Phylogenetic tree of *Macrotis lagotis* and *Macrotis leucura*. Support values for major branches are given as bootstrap support values (in black as a percentage) from the maximum likelihood analysis and as posterior probabilities (in blue) from the Bayesian analysis. Scale bar indicates 0.03 substitutions per site. Divergence times for the bilby–bandicoot and Ninu–Yallara divergences are provided in millions of years (Myr). The alpha-numeric text corresponds to the sample names (Table S1) and green and orange text represents the semi-arid and temperate Ninu, respectively. **d**, PCA plot of Yallara (yellow), the semi-arid Ninu (green) and temperate Ninu (orange) male whole genome resequence (WGR) data. NT, Northern Territory; QLD, Queensland; SA, South Australia; PC, principal component; WA, Western Australia.

due to the species' decline is a recognized threat to the persistence of bilbies in the landscape[2].

Historically Ninu (greater bilby) were wide ranging, distributed across both arid and temperate regions, while the Yallara (lesser bilby) were restricted to the sandy deserts[3]. The declines of both bilby species are attributed to the introduction of feral pests into Australia by European settlers, particularly predation by cats (*Felis catus*) and foxes (*Vulpes vulpes*), competition from European rabbits (*Oryctolagus cuniculus*), as well as changes to cultural fire regimes[4]. Sadly, the Yallara is now extinct, last reported alive in 1931, although it may have survived in some desert areas until the 1960s[4] and was well known to the Indigenous peoples of the central deserts (Fig. 1b)[5]. The now threatened Ninu is believed to exist in only 20% of its former range in the semi-arid regions of north Western Australia (WA), the Northern Territory (NT) and Queensland (QLD) (Fig. 1a)[6]. Therefore, the conservation of the Ninu is now of critical importance as it is the last extant member of the Thylacomyidae marsupial family.

To ensure the long-term survival of the species, Ninu are managed as a metapopulation consisting of individuals in zoos as well as those in several fenced sanctuaries and on islands. Although Ninu were periodically held in zoos in the early 20th century, a captive breeding colony was formally established in 1979. The zoo-based populations were

managed as two separate evolutionary units (NT/WA and QLD), until 2016 when they were combined into one metapopulation[7], resulting in the mixing of different bilby bloodlines from many Indigenous communities making it difficult to attribute metapopulation individuals to any particular traditional owner group. Since 1996 Ninu from the zoo population have been released to large, fenced sanctuaries and islands, noting that the concept of translocations is culturally sensitive to many Indigenous communities. The genetic consequences of these ongoing translocation events are not known but are explored here. Current methods used to understand the status of wild Ninu populations rely on a combination of track and scat surveys often undertaken by Indigenous rangers[8], including microsatellite genotyping of scats to estimate abundance[9]. Although these surveys ascertain Ninu presence/absence and breeding (based on expert interpretation of the size of tracks and scats), they provide no insight into sex ratios, relatedness or gene flow between wild locations. In remote areas of the Ninu range, many Indigenous communities follow traditional practices for actively managing fire regimes and invasive species[1,10]. In this Article, we develop genetic tools to support Indigenous communities in conserving not only the Ninu, but the cultural practices associated with the species. We also for the first time undertake a genetic comparison between the managed metapopulation and wild individuals to better understand if the metapopulation is representative of wild genetic diversity to ensure we can manage its long-term adaptive potential.

Bilbies have several unique biological features that could be better understood using comparative genomics. Characterized by their large ears, bilbies are burrowing, nocturnal omnivores that have strong forelimbs with long claws for digging and finding food[11] (Fig. 1a). Bilbies are physiologically well adapted to arid environments, having low metabolic rates and low water turnover[11]. They do not drink free water but are able to obtain sufficient moisture from their food[12], which consists of insects, insect larvae, seeds, bulbs and fungi[13,14]. Although not the largest of marsupials, Ninu males (50–84 cm; 0.66–2.5 kg) and females (49–68 cm; 0.66–1.1 kg)[4] are proportionally bigger at birth than other species, which has been attributed to their complex placenta and broad milk composition[13]. Bilbies, along with bandicoots, belong to the order Peramelemorphia, and unlike other marsupials, have a short-lived chorioallantoic placenta as well as a choriovitelline placenta[15]. Like other digging marsupials, females have a backward-facing pouch and have one or two (rarely three) offspring per breeding event (average 1.48–1.94)[16]. Females are polyoestrous with an oestrous cycle of 12–37 days (20.6 ± 7.3 days; $N = 14$) and an oestrous duration of 2–11 days (4.3 ± 2.1 days)[15]. Gestation is for 14 ± 1.4 days ($N = 4$) and offspring exit the pouch at around 80 ± 2 days ($N = 6$) with a lactation period of 90 days[13]. Females are sexually mature from 5 months of age and males from 7 months[16]. Owing to the extreme Australian climate, characterized by periods of extensive drought followed by flooding rains, bilbies are known as a boom–bust species[13]. That is, their breeding seasons depend on food availability and rainfall, meaning population numbers have the capacity to expand and contract rapidly in relation to climatic patterns. In times of high resource availability, they can breed every 3 months, producing up to seven offspring in a 12-month period[15]; fecundity decreases in times of low resource availability.

Here, we present the first chromosome length reference genome assembly for this threatened elusive species, the Ninu. This comprehensive study arose from the need to understand the population viability of both the Ninu metapopulation, as well as wild individuals managed by the Kiwirrkurra Indigenous rangers. We have achieved this by generating a reference genome to develop and test an innovative scat genotyping tool to inform current and future conservation actions and undertake ecological assessments, in addition to understanding the current genetic status of the Ninu metapopulation. Using resequenced genomes from both the Ninu and Yallara, we have assembled the extinct Yallara genome and greatly increased our knowledge of their unique biology and demographic histories.

## The genomic landscape

The Ninu reference genome is now one of the highest-quality marsupial genomes so far, comparable with that of the koala[17], offering insights into biology, evolution and contemporary population dynamics. The female Ninu reference genome was generated using a combination of long-read sequencing (HiFi), HiC (Omni-C) scaffolding and short-read (Illumina) polishing. The assembly is 3.66 Gb in size, which is larger than other marsupial genomes (Extended Data Table 2), with 95.6% assigned to nine nuclear chromosome scaffolds and the mitochondrial genome (scaffold N50, 343.85 Mb; 0.34% gaps and 93.5% complete mammalian benchmarking universal single-copy orthologues (BUSCO); Extended Data Table 2 and Extended Data Fig. 1). Chromosome 1 is extremely large at 934,426,298 bp. The global transcriptome (including non-coding transcripts) contains 39,106 genes, with an average transcript length of 6,833 bp and an N50 of 13.4 kb (Extended Data Table 3). For all protein-coding transcripts, the longest open reading frame had an average transcript length of 1,010 bp and N50 of 1,620 bp. 47.45% of the genome is masked as repetitive (Extended Data Table 3). A Yallara genome assembly was generated from a skull sample collected in 1898 and sequenced with short-read (Illumina) sequencing (male NMVC7087; Supplementary Table 1). We used the Burrows-Wheeler Aligner (BWA) aln algorithm to align reads to the Ninu genome (version 1.9; Supplementary Section 1.7) resulting in a Yallara genome assembly that is 3.50 Gb in size (6,329,012 contigs; 19.74% gaps; 75.2% complete mammalian BUSCO; Extended Data Table 2).

A total of 12 Ninu from the metapopulation were used for the whole genome resequencing (WGR) by Illumina Novaseq with an average coverage of 23.9 ± 4.0× (± standard deviation (s.d.); range 13.8–29.6×; Supplementary Table 1). Six individuals (three males and three females) were from a temperate island (35° S, 136° E) and the other six (three males and three females) were from a semi-arid region (26° S, 146° E; Fig. 1a). DNA was extracted from four male and one female Yallara samples collected between 1895 and 1931 (Supplementary Table 1). Four of these were resequenced using Illumina Novaseq, with 6.0 ± 6.1× mean coverage (range: 0.73–12.82×). The phylogenetic relationship among the Ninu and Yallara individuals was confirmed using both full mitogenomes and whole genome nuclear data. Mitogenomes were extracted from the final BAM files generated for the high coverage Ninu ($N = 6$) and Yallara ($N = 2$) individuals (Supplementary Table 1), and a phylogenetic tree was constructed using both maximum likelihood and Bayesian methods (Fig. 1c). Principal component analyses (PCAs) were also generated for the whole genome datasets, which both confirmed the mitogenome divergence results (Fig. 1d, Supplementary Note 1.7 and Supplementary Fig. 7). For the dataset including only high-coverage individuals (total 2,787 variants), principal component 1 splits the Yallara and Ninu explaining 61.6% of the variance (Fig. 1d). Principal component 2 splits the Ninu samples into semi-arid and temperate samples, explaining a further 11% of the variance observed (Fig. 1d) consistent with the mitogenome results (Fig. 1c).

## Genome-informed conservation

Our analyses of the Ninu and Yallara effective population sizes, using pairwise and multiple sequentially Markovian coalescent analyses (PSMC and MSMC), revealed initial declines at 500,000–8,000,000 years ago and 300,000–4,000,000 years ago, respectively (Fig. 2a). Both contractions coincide with the cooling of the global surface temperature before the last glacial period. The Ninu population expanded 100,000–500,000 years ago, followed by a possible decline in the last 100,000 years. However, this pattern is not well resolved as the bootstrap replicates lacked a signal for a population contraction and the pattern lies at the limits of sequentially Markovian coalescent analyses (SMC) estimation. There is no clear separation between the semi-arid and temperate populations during the timeframe of inference (Fig. 2b). The effective population size of Yallara may be underestimated due to the low coverage (mean 6.0×), falling below previously

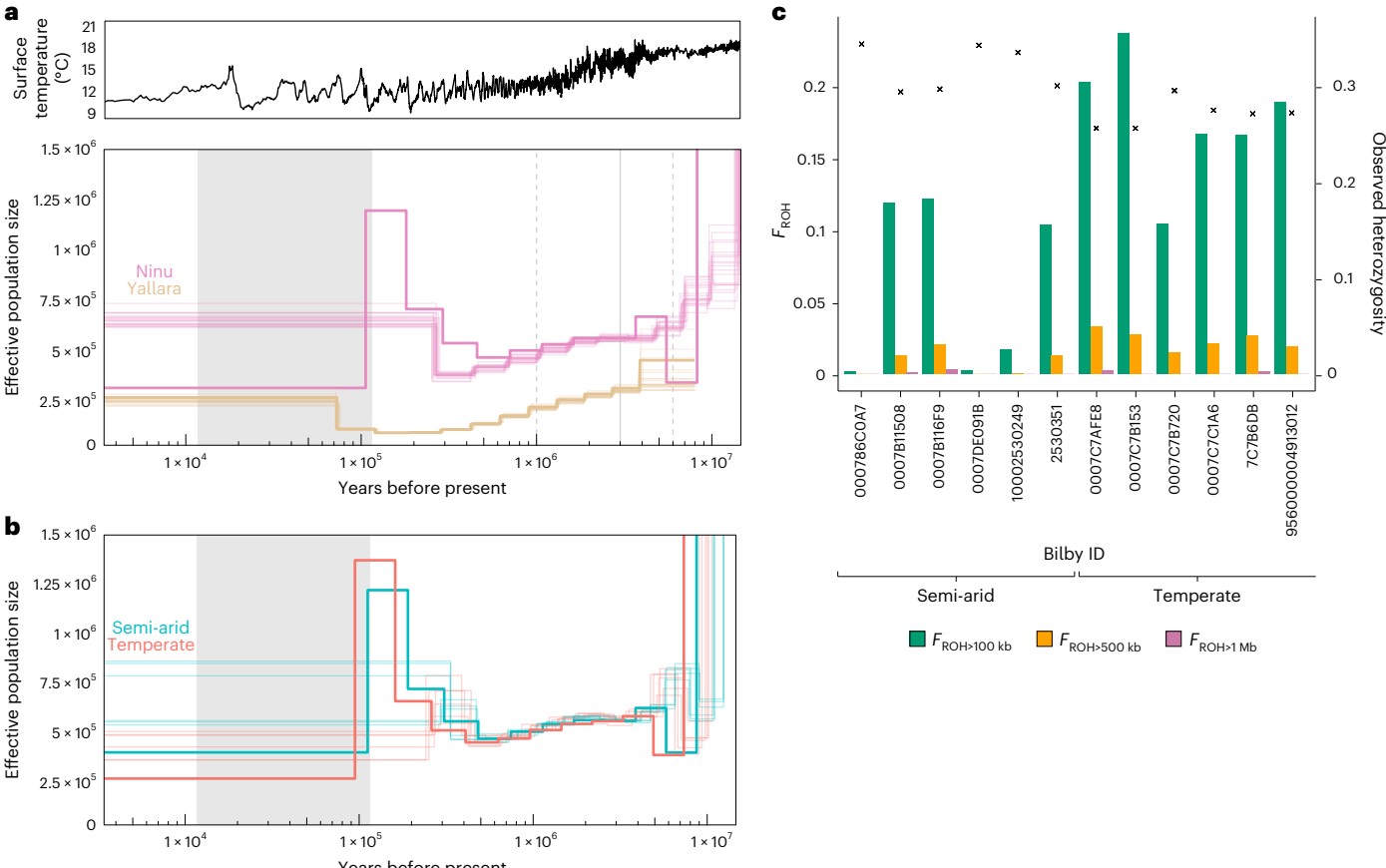

**Fig. 2 | Demographic histories of both the Ninu and Yallara and inbreeding and heterozygosity statistics from the Ninu resequenced genomes. a**, Top: surface temperature (°C) based on a five-point running mean of $\delta^{18}$O data[125]. Bottom: effective population size through time estimated from Ninu ($n$ = 4; pink) and Yallara ($n$ = 2; beige) scaled using a Tasmanian devil mutation rate ($1.17 \times 10^{-9}$ per nucleotide site per generation) and a generation time of 2 years[88]. The bold coloured lines indicate the combined MSMC estimations per species, with 20 bootstrap replicates (thin lines). The grey area indicates the last glacial period. The solid grey vertical line indicates the mean estimated divergence time between greater and lesser bilbies approximately 3 million years ago, with dashed

vertical lines indicating the 95% credible intervals[126]. **b**, Effective population size through time estimated from Ninu samples from temperate (red) and semi-arid (teal) regions. For each group of Ninu, the bold line indicates the combined MSMC estimation using the four individuals with the highest mean sequencing coverage (Supplementary Table 1). Thin lines indicate population size histories inferred from individual genomes with PSMC. Grey shading indicates the last glacial period. **c**, $F_{ROH}$ and observed heterozygosity for the 12 resequenced Ninu. The coloured bars represent the proportion of the genome in ROH >100 kb, ROH >500 kb and ROH >1 Mb, while the crosses represent observed heterozygosity.

recommended thresholds for SMC analyses[18,19]. Similarly, the flattened peaks and slight offset between the two MSMC estimations of temperate and semi-arid Ninu may be an outcome of differences in the mean sequencing coverage, where semi-arid individuals generally had higher genome coverage (Supplementary Table 1).

Investigation of the 12 resequenced Ninu genomes reveals differences in heterozygosity and ROH. As expected from the significant inbreeding observed in the reduced representation sequencing data (RRS; Table 1), the six Ninu from the temperate island population (Thistle Island) had lower heterozygosity and generally higher runs of homozygosity (ROH)-based inbreeding coefficients ($F_{ROH}$) than those from the semi-arid population (Fig. 2c and Supplementary Table 2), even though unrelated individuals were selected for resequencing. These results are probably due to the relatively small founder size that started the island population in the temperate region ($N$ = 21) and limited gene flow with other populations for ~35–40 generations (Supplementary Note 2.3). $F_{ROH}$ values of semi-arid Ninu were variable, but generally comprised fewer short ROHs than the temperate individuals (Fig. 2c and Supplementary Note 2.2).

The semi-arid Ninu population was originally sourced from the last remaining Ninu population at Astrebla Downs, QLD (24.20° S, 140.55° E; Fig. 1a) and have only spent ~1–7 generations in captivity. The number of

short ROHs in some of the semi-arid individuals suggests past inbreeding in the wild populations, probably caused by declining population sizes. The long ROHs in these individuals suggest recent inbreeding, potentially within the captive population, indicating management strategies that avoid inbreeding and high population relatedness in the metapopulation should continue. In general, Ninu have relatively few long ROH compared with other threatened mammals[20–22]. The fewer long ROHs in Ninu, despite their history of small founder sizes and captive breeding, could be partially attributed to their boom–bust demographic history and/or their shorter generation time (hence higher per-year recombination rate) and potentially higher substitution rate than those of larger mammals[23]. Further work could tease apart the influence of the species' demography and intrinsic biological characteristics on the ROH distribution.

Efforts to improve Ninu genetic diversity through genetically driven population management actions were successful. The expansion of the managed metapopulation occurred between 2016 and 2021, where existing zoo-based and fenced sanctuary populations were used as source populations for new fenced sanctuaries (Fig. 3a and Supplementary Notes 2.3 and 2.4). We used over 9,000 single-nucleotide polymorphisms (SNPs) called from RRS (Table 1) of 363 individuals aligned to the reference genome to inform translocations and understand

**Table 1 | Population genetic statistics, excluding Birdsville (QLD) and Currawinya (wild) samples (*n*=1 for each population)**

| Population | Samples | N loci genotyped | $H_O$ (s.e.m.) | $H_E$ (s.e.m.) | AR* (s.e.m.) | $F_{IS}$ (95% CI) | MK (s.e.m.) | $N_E$ (95% CI) | Harmonic mean N |
|---|---|---|---|---|---|---|---|---|---|
| Group 1—source populations | | | | | | | | | |
| Arid Recovery | 16 | 9,855 | 0.1700 (0.002071) | 0.1795 (0.001928) | 1.179 (0.001930) | 0.0369 (−0.0011, 0.0667) | 0.0850 (0.0023) | 207.1 (77.1, INF) | 12.7 |
| Kimberley (wild) | 5 | 9,835 | 0.1451 (0.002331) | 0.1570 (0.002159) | 1.156 (0.002160) | −0.0082 (−0.1168, 0.0660) | 0.1358 (0.0125) | INF (10.4, INF) | 4.4 |
| Pilbara (wild) | 5 | 9,749 | 0.1005 (0.002079) | 0.1705 (0.002602) | 1.156 (0.002392) | 0.3007 (0.0840, 0.8603) | 0.0747 (0.0067) | INF (INF, INF) | 3.0 |
| Scotia | 53 | 9,905 | 0.1499 (0.001855) | 0.1690 (0.001818) | 1.169 (0.001814) | 0.0981 (0.0442, 0.1393) | 0.1268 (0.0020) | 5.1 (3.3, 6.6) | 29.0 |
| Thistle Island | 89 | 9,905 | 0.1761 (0.001909) | 0.1835 (0.001828) | 1.183 (0.001827) | 0.0344 (0.0229, 0.0462) | 0.0626 (0.0005) | 83.8 (63.8, 106.7) | 72.3 |
| Venus Bay | 5 | 9,807 | 0.1319 (0.002460) | 0.1317 (0.002090) | 1.131 (0.002098) | −0.0374 (−0.2736, 0.1793) | 0.2443 (0.0312) | 4.7 (0.6, INF) | 4.7 |
| Yookamurra 1 | 19 | 9,621 | 0.1245 (0.002433) | 0.1062 (0.001889) | 1.113 (0.002074) | 0.0599 (−0.0060, 0.1527) | 0.2644 (0.0052) | 8.7 (2.8, 34.7) | 8.5 |
| Yookamurra 2 | 3 | 9,792 | 0.1546 (0.002727) | 0.1554 (0.002370) | 1.155 (0.002359) | −0.1151 (−0.1773, −0.0386) | 0.1681 (0.0127) | INF (0.2, INF) | 2.9 |
| ZAA | 78 | 9,906 | 0.1839 (0.001707) | 0.1964 (0.001612) | 1.196 (0.001611) | 0.0556 (0.0352, 0.0752) | 0.0376 (0.0010) | 20.8 (17.4, 25.1) | 58.4 |
| Group 2—translocated population founders | | | | | | | | | |
| Currawinya | 35 | 9,906 | 0.1837 (0.001774) | 0.1978 (0.001672) | 1.198 (0.001669) | – | 0.0421 (0.0025) | – | – |
| Dubbo | 18 | 9,906 | 0.1916 (0.001878) | 0.2024 (0.001730) | 1.202 (0.001722) | – | 0.0355 (0.0044) | – | – |
| Mallee Cliffs | 50 | 9,904 | 0.1709 (0.001651) | 0.1981 (0.001661) | 1.198 (0.001658) | – | 0.0377 (0.0014) | – | – |
| Mt Gibson | 26 | 9,904 | 0.1470 (0.001871) | 0.1710 (0.001846) | 1.170 (0.001836) | – | 0.0948 (0.0047) | – | – |
| Pilliga | 36 | 9,900 | 0.1752 (0.001842) | 0.1917 (0.001756) | 1.191 (0.001755) | – | 0.0527 (0.0018) | – | – |
| Group 3—offspring populations | | | | | | | | | |
| Currawinya | 35 | 9,906 | 0.1913 (0.001927) | 0.1925 (0.001724) | 1.192 (0.001724) | −0.0127 (−0.0391, 0.0141) | 0.0275 (0.0022) | 16.5 (13.4, 26.6) | 25.7 |
| Dubbo | 46 | 9,905 | 0.1888 (0.001972) | 0.1873 (0.001743) | 1.187 (0.001744) | −0.0228 (−0.0478, −0.0029) | 0.0342 (0.0019) | 13.7 (10.7, 18.5) | 33.3 |

Standard error of the mean (s.e.m.) is calculated as s.d. divided by√(N), where N is the number of genotyped loci for each population. The full set of 9,906 SNPs was used. Effective population size ($N_E$) was calculated on randomly selected subset of 5,000 loci and reported as the estimated $N_E$ (no singletons), jackknifed 95% CIs and harmonic mean sample size. As group 2 represents the founding animals that were sourced from group 1 populations, inbreeding coefficient ($F_{IS}$) and $N_E$ were not calculated as the Wahlund effect would likely influence results due to the mixing of diverse source populations at translocated sites[124]. AR, allelic richness; CI, confidence interval; $H_E$, expected heterozygosity; $H_O$, observed heterozygosity; MK, mean kinship. *Rarefied to 2.

the genetic outcomes of our management recommendations (Supplementary Note 2.6). Observed heterozygosity across these source populations ranged from 0.1005 to 0.1839 (Fig. 3c and Table 1). As expected, the mixed translocated populations had higher observed heterozygosity ranging from 0.1470 to 0.1916, and this flowed through to the two offspring populations, assessed as part of this study, with 0.1888 and 0.1913 (Fig. 3c and Table 1). Most populations exhibited lower observed heterozygosity than expected under Hardy Weinberg equilibrium (Table 1), indicating that inbreeding may be occurring.

In concordance with the observed excess of homozygosity, mean inbreeding ($F_{IS}$) of the source populations was statistically significant for

the Pilbara, Scotia, Thistle Island and the zoo-based (ZAA) populations (Table 1). High allelic richness was observed across the translocated populations, probably due to these sites being recently established by genetically differentiated source populations (Supplementary Fig. 10). Mean kinship (relatedness) was highest in some of the isolated source populations (Venus Bay, Yookamurra 1 and Yookamurra 2). Effective population size estimates were low across the populations and were generally estimated with poor precision (Table 1). It is important to note that sample size can affect the accuracy of estimating such population genetic statistics, so results from populations with low sample sizes (for example, fewer than six individuals) should be treated with caution[24].

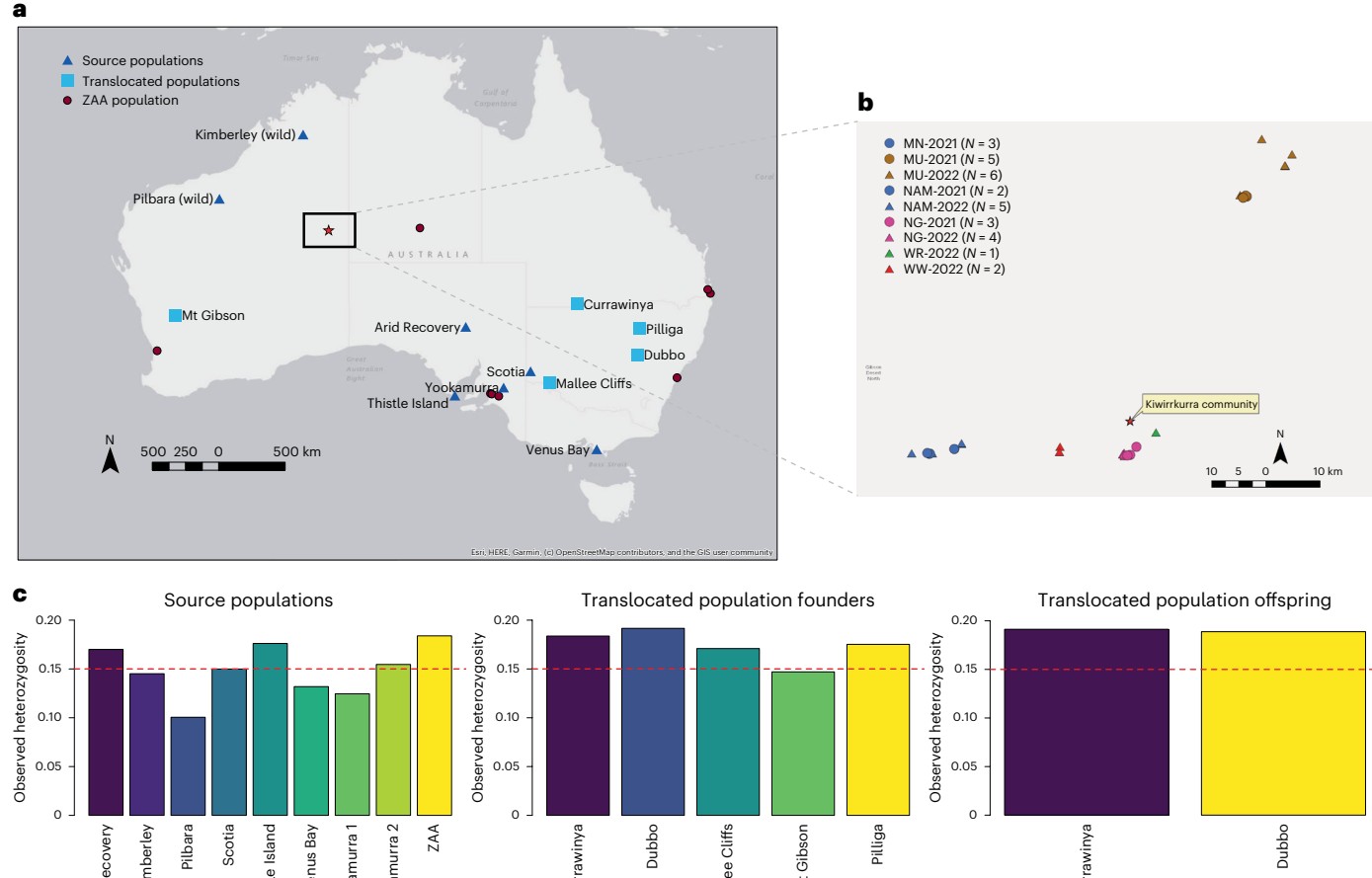

**Fig. 3 | Wild and metapopulation sampling locations and observed heterozygosity of the source and translocated populations. a**, Map showing all sampling locations and the Kiwirrkurra Community (red star). **b**, Inset map from larger map (black box in **a**) depicting the locations of the unique Ninu identified from the scat samples collected from Ninu colonies south and north-east of the Kiwirrkurra Community. **c**, The observed heterozygosity of each of these populations. The dashed red line indicates $H_o$ of 0.15 to show increases in heterozygosity as a result of the translocation programme. Maps made in ArcMap v10.7 powered by Environmental Systems Research Institute.

A toolkit for measuring the success of conservation efforts has been developed and tested. Using RRS SNPs aligned to the reference genome, a MassARRAY (mass spectrometry with end-point polymerase chain reaction) panel of 35 autosomal and four sex-linked markers was developed (Supplementary Note 2.5). SNP loci were selected on the basis of high minor allele frequency (>0.30) in populations across the species distribution and with high reproducibility. This was used to genotype 195 scats collected by Indigenous rangers from the Kiwirrkurra Community across two locations, south and north-east of the community, in 2021 and 2022 (Fig. 3b). Traditional hunting of feral cats and regular fire management is implemented in areas south of the Kiwirrkurra Community to reduce predation pressure but not in the north-eastern Ninu colonies. Indigenous rangers were interested in determining baseline data on the abundance of Ninu in the two areas before predator baiting is instigated around the northern sites, and whether these two colonies have become genetically isolated. Whilst survey of the north-eastern Ninu colonies was not as comprehensive as the southern colonies, we detected more Ninu ($N = 16$) in the area where traditional cat hunting occurs than in the north-eastern area ($N = 9$; Fig. 3b and Supplementary Table 3). Cumulatively, based on the 35 autosomal SNPs from the MassARRAY panel, the genetic diversity of the wild Ninu population at Kiwirrkurra was comparable with other wild populations in the Pilbara and Kimberley ($H_o = 0.34$, 0.29 and 0.37, respectively; Extended Data Table 4). The north-eastern and southern Ninu colonies are located approximately 70 km apart but appear to be connected with detection of several half-sibling and higher relationships amongst individuals located in the north-eastern colony and southern colonies, and little genetic structuring observed in principal coordinate analysis (Supplementary Fig. 1).

Managing metapopulations is complex but has been assisted greatly in recent years with genetic data[25]. There is evidence of population stratification based on the demographic and translocation history of the Ninu populations (Supplementary Fig. 2 and Supplementary Fig. 10)[26]. Throughout this study a combination of the known translocation histories and genetic data were used to develop translocation recommendations for the National Bilby Recovery Team (Supplementary Note 2.6). These recommendations included maximizing genetic diversity and value of the metapopulation, sourcing wild bilbies, biobanking genetic samples and using the scat method to undertake a nationwide survey (Supplementary Note 2.6). Our recommendations resulted in the movement of 225 individuals between 2016 and 2021 to establish five new populations within sanctuary areas (Fig. 3a and Supplementary Note 2.3). Offspring sampled at two of these newly established locations show the benefits of genetic mixing within the metapopulation (Fig. 3c and Table 1), suggesting that this practice should continue to maintain Ninu genetic diversity in a protected environment. Based on the success of the scat case study, Indigenous rangers and other conservation agencies can now use this method to undertake a whole country survey for the species to better understand its distribution and movement between isolated

populations, both in the wild and sanctuary locations, and estimate census population size[27].

## Unique biological insights from the Ninu genome data

Further to our primary aim of using genetic data to assess and inform current and future conservation management, we used the resources generated in this study alongside comparative genomics approaches to explore the genomic basis of the Ninu's unique adaptations.

### Differences between semi-arid and temperate individuals

To assess potential adaptive allele frequency differences between semi-arid and temperate individuals, we performed a genome-wide association study (GWAS) using the 12 resequenced Ninu genomes. A total of 3,858 SNPs that met our criteria (bi-allelic SNPs, no missing data and minor allele frequency >0.05) were common across all three association analyses (Chi-squared association, Fisher's test and $F_{ST}$ outlier test). As a result, we identified 339 enriched genes between semi-arid and temperate individuals (Extended Data Fig. 2; a full list of Gene Ontology (GO) terms and associated genes is given in Supplementary Table 4). As only two individuals had high sequencing coverage for the Yallara (Supplementary Table 1), association analyses could not be undertaken.

### Metabolism and olfactory receptors

Ninu have the lowest standard metabolic rate and the largest olfactory bulbs of any marsupial, which is reflected in their genome. The top ten GO terms between temperate and semi-arid Ninu are associated with genes involved in anatomical structure (including *SYNE1* and *FMR1* involved in brain development), metabolic and cellular pathways (including *BBOX1* and *ACSBG1* involved in fatty acid metabolism) and response to stress (including *GRM7* involved in neurotransmission in mammalian central nervous systems; Extended Data Fig. 2 and Supplementary Table 4). It is not surprising that seven of the top ten GO terms are involved in cell differentiation, transport and metabolic pathways as Ninu are known to have a low standard metabolic rate (58% of eutherian standard) compared with other marsupials (70% of eutherian standard)[13], as well as consuming very small quantities of water (56–68 ml per day) compared with bandicoots (46–341 ml per day), koala (296–414 ml per day) and possums (114–140 ml per day)[28].

Interestingly, the gene families involved in anatomical structure development, a range of metabolic processes and response to stress were also identified as fast evolving using a Computational Analysis of gene Family Evolution (CAFE) of Ninu compared with nine other species' genomes (Extended Data Fig. 3, Supplementary Table 5 and Supplementary Note 3), including five marsupials across the marsupial lineage (brown antechinus, Tasmanian devil, koala, Tammar wallaby and opossum), one monotreme (platypus) and three eutherian mammals (human, mouse and cow). We also show Ninu to have the highest number of annotated olfactory receptor genes (*OR1D2* and *OR1D5*) across these ten species (Extended Data Table 5 and Supplementary Note 3.7). This is unsurprising as bilbies rely on olfactory cues for locating food, leaving scent markings for male–male signalling[29] and avoiding predators[30,31], and have larger olfactory bulbs than other marsupials of similar body size[32].

### Genomics of the reproductive and immune systems

The association analyses of the male Ninu from the temperate and semi-arid populations revealed four genes expressed in the testis (*SPEF2*, *TBC1D21*, *SYNE1* and *NME8*) that are involved in spermatogenesis, with each population having private and fixed alleles for all four genes. *SPEF2* is critical in sperm tail development and head shape[33]; *TBC1D21* is similarly essential for sperm tail function[34]. *NME8* is involved in sperm tail maturation[35], while *SYNE1* (*KASH1*) is involved in sperm head formation[36]. It is tempting to speculate on the functional effect

of these fixed differences in genes essential for male fertility between these two populations. However, the small testis size of Ninu relative to body mass[37] and the fact that only litters with single paternity have been observed[38], suggests that they do not have a multi-male mating system that might produce differential rates of sperm competition between different populations. Instead, these sequence differences may be due to either population genetic differences caused by drift, or possibly higher mutation rates in animals living in warmer climates. Future studies in this area should examine mating structures, differential testis gene expression, sperm function and dominance in male breeding activity across Ninu populations.

The Peramelemorphia (bilbies and bandicoots) standout amongst marsupials due to their invasive chorioallantoic placenta, while most other marsupials rely on a yolk sac (choriovitelline) placenta[39]. All 115 of the genes that show conserved chorioallantoic expression across all eutherians[40] are expressed in the Ninu uterus (transcript per million >2). During formation of the peramelemorphian chorioallantoic placenta, uterine epithelia and trophoblast cells fuse together to form a heterokaryotic syncytium[41]. Syncytia in both eutherians and marsupials have evolved through the incorporation of fusogenic retro-viral envelope (env) genes referred to as syncytins[42,43]. All marsupials examined have at least one incorporated syncytin (*Env2*)[43]. Since a chorioallantoic syncytium is unique to Peramelemorphia, we might expect that incorporation of multiple syncytins has occurred in this group. The Ninu genome has a diversity of encoded retro-elements, including more than 45,000 long terminal repeat retrotransposons, from which further envelope genes could have been co-opted. This genome provides the foundation for future studies of the fusion of maternal and foetal cells in the unique peramelemorphian placenta, if placental tissues can be obtained.

The immune gene repertoire of the Ninu is similar to those of other marsupials[44,45], with marsupial-specific genes and eutherian orthologues identified. Immune genes were annotated in the Ninu genome and transcriptome using similarity-based search methods such as BLAST[46] and HMMER[47] with known marsupial immune gene sequences as queries. This resulted in the manual characterization of over 562 immune genes, from six immune gene families or groups (Extended Data Table 6): cytokines, toll-like receptors (TLR), the major histocompatibility complex (MHC-I, MHC-II and MHC-III), natural killer cell (NK) receptors, immunoglobulins (Ig) and T cell receptors (TCR). Relatively conserved immune genes, such as TLRs and constant regions of TCR and Ig, were identified in addition to those immune genes unique to the marsupial lineage; including *TLR1/6*, *TCRμ*, MHC-I (*-UM*) and MHC-II (*-DA*, *-DB* and *-DC*) genes (Supplementary Fig. 3). Large marsupial-specific gene expansions within the LRC NK receptors were characterized, as well as the reduced gene content within the NKC cluster of NK receptors (Extended Data Table 6). Consistent with other marsupials investigated so far, Igδ was not found[48]. The organization of the MHC region in the Ninu is similar to those of other marsupials in that the MHC-I and -II genes are interspersed; there is a MHC-III region and framework region, and the core MHC cluster is flanked by extended MHC genes (Extended Data Fig. 4)[49]. However, a few key distinct differences exist, with the four *DAB* genes positioned 8.7 Mb from the flanking extended region and the translocation of four MHC-I genes onto scaffold 1 present (Extended Data Fig. 4). The mean sequence similarity between MHC-I genes was 76.4% in coding sequences and 63.6% in the translated amino acids (Supplementary Table 6). The MHC-I genes that translocated onto scaffold 1 show very high sequence similarities (for example, 99.3% between *-UA* and *-UB*, and 99% between *-UC* and *-UD*) and strong bootstrap support (100%).

Interestingly, there were fewer MHC-I and Ig variable genes in the Ninu, Tasmanian devil and opossum than in the other marsupials (Extended Data Table 6). The loss of MHC-I and Ig variable genes in the Ninu may be due to its invasive placenta, placing embryonic tissues in closer proximity to maternal tissues compared with epitheliochorial placentation. In the opossum, the brief phase of placental attachment

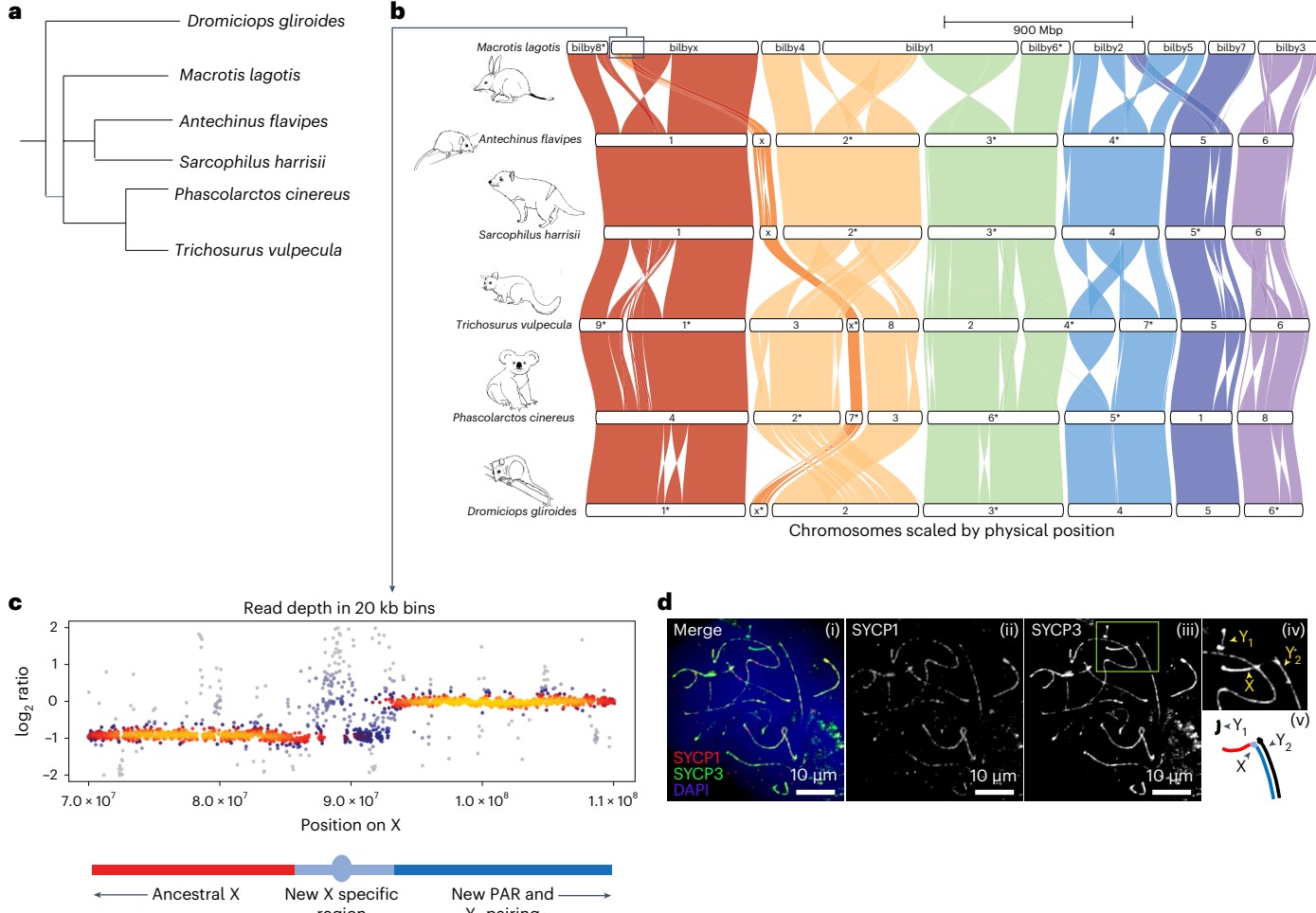

**Fig. 4 | Synteny map showing differences in marsupial chromosomes including the bilby XY₁Y₂ chromosome. a**, Indicative phylogeny of marsupial species included in the synteny map. **b**, Genespace synteny plots. Coloured blocks represent chromosome scaffolds for each species. Chromosomes are ordered to maximize visual synteny relative to neighbouring genomes, with the Ninu chromosomes defining the starting order and orientation. Each chromosome is labelled with the species and chromosome number (or X), with an asterisk when reversed to minimize inversions. **c**, Heatscatter of read depth ratios (relative to the pseudo-autosomal region (PAR) mean) of male

genome sequence data in 20 kb bins of the compound X, demarcated into the ancestral X at half read depth, new X-specific region and a large PAR with full read depth that pairs with the Y₂ during male meiosis. **d**, Representative image of Ninu primary spermatocyte stained by immunofluorescence with DAPI, SYCP1 and SYCP3 (i–iii), along with an inset depicting sex chromosomes (iv) and a model of sex chromosome pairing (v). The arrows indicate the position of sex chromosomes, with colours matching **c**. Scale bar, 10 μm/2 μm. The experiment was repeated twice.

is accompanied by inflammatory signalling[50], although there is little evidence for maternal recognition in marsupials outside of the macropods[51]. Changes in vertebrate immunity have been noted in other species where pregnancy has evolved including the losses/modifications to the MHC-II pathway and expansion of the MHC-I repertoire in seahorses and pipefish[52]. However, without a Ninu pregnant uterus or placenta, we can only speculate on the meaning of the loss of MHC-I and Ig variable genes in this species relative to the presence of 115 eutherian conserved chorioallantoic genes.

### Bilby chromosomes

As with several other marsupial species, Ninu chromosomes have a number of rearrangements (Fig. 4a,b and Supplementary Fig. 4). The Ninu genome provides insights into chromosome evolution showing the Ninu XY₁Y₂ system was generated by fusion of the X with the long arm of an autosome. Early work on marsupial karyotypes described a 2*n* = 18 complement with nine chromosome pairs in the Ninu, including a large submetacentric X in females, and a 2*n* = 19 complement in males with a single X, and two male-specific Y chromosomes[53]. The Y₁ chromosome

is very small, as is the case with the Y in most marsupial species, and represents the ancestral Y. In contrast, the Y₂ is a long telocentric chromosome, with a size and morphology like that of the long arm (Xq) of the Ninu X[53]. However, alignment of the Illumina male genome sequence to the female Ninu reference genome now reveals more detail of this XY₁Y₂ system. Read depth analysis shows that the compound X is demarcated into an X-specific region (Xp) with half read depth, and a large pseudo-autosomal region (Xq) with full read depth in males that pairs with Y₂ at male meiosis (Fig. 4c). Additionally, reduced read depth extends into the fused autosome, so represents new X-specific material. Interestingly, this region does not pair with Y₂ during male meiotic prophase I (Fig. 4d). Future work on bilby Y₂ chromosomes will yield information about sex chromosome differentiation and future comparisons between male and female transcriptomes will inform our understanding of meiotic sex chromosome inactivation in marsupials.

### Conclusions

Rapid advances in genome sequencing technology have allowed us to sequence genomes for both the extant greater bilby (Ninu) and the

extinct Yallara leading to advances in our understanding of their unique biology and a toolkit for measuring the success of conservation efforts. Ninu are the only surviving species in their marsupial family, are known ecosystem engineers and have ongoing important cultural value to Indigenous Australians. Here, we showcase the bilby's unusual biology, in addition to their cultural value and importance. We recognize the many First Nations names for the species (Extended Data Table 1) and use the name Ninu in recognition of the wild samples provided by the Kiwirrkurra Indigenous rangers.

Although once wide ranging across the continent, the long-term survival of Ninu in the wild is hampered by the presence of invasive pest species and altered fire regimes. Management of wild populations so far has been hindered by their cryptic, nocturnal nature making our scat genotyping array panel a critical component for understanding wild populations in the future due to the relative ease of scat collection and low cost compared with trapping and tissue sampling. For the first time, we provide an understanding of the remaining wild Ninu genetic diversity relative to the managed semi-wild populations that are currently a conservation reservoir for the species. Next steps for the scat protocol are to continue to use it for the bilby monitoring programme in the Pilbara and undertake a nationwide scat survey in collaboration with Indigenous communities, not-for-profit organizations and Indigenous school groups to provide a more comprehensive comparison between the metapopulation and wild remnant populations. Throughout our project, we have worked closely with conservation managers and Indigenous rangers, so our latest genetic research data has informed their management actions in real-time. Notably, we have provided a worked example of the value of a high-quality reference genome, through gene discovery and interpretation, to downstream applied conservation actions. Our approach showcases what can be achieved when academics partner with Indigenous communities to understand culturally and ecologically important species and is relevant to genome biologists, evolutionary and conservation geneticists, and conservation managers. This is not just another genome project but rather represents the holistic value of reference genomes to answer key Indigenous and conservation end-user management questions and understand the evolution and fundamental biology of a unique Australian species.

## Methods

A full description of the methods can be found in Supplementary Notes.

**Genome sequencing and assembly of the Ninu reference genome.**
Samples were collected opportunistically following medical euthanasia of a female Ninu at Perth Zoo (2018). DNA was used to assemble a high-quality reference genome, using a hybrid approach of 10x Genomics linked-read sequencing[54], Pacific Biosciences (PacBio) HiFi sequencing[55] and Dovetail Omni-C. For 10x Genomics linked-read sequencing, high molecular weight DNA was extracted from 25 mg of spleen using the MagAttract HMW DNA kit (Qiagen catalogue 67563) and sequenced on a NovaSeq 6000 S1 flowcell (Illumina) using 150 bp paired-end reads and obtaining ~57× coverage. For the HiFi sequencing, high molecular weight DNA was extracted from 100 mg kidney using the Nanobind tissue big DNA kit (Circulomics catalogue NB-900-701-01) and sequenced using two SMRT cells of the PacBio Sequel II in circular consensus mode obtaining ~10× coverage. For HiC sequencing, 20 mg of ground flash-frozen spleen tissue was input into the Dovetail Genomics Omni-C proximity ligation assay (version 1.3), with a modified 1:10 dilution to the digestion enzyme. The proximity ligated DNA was split at the end of stage 3 into two 150 ng aliquots and taken through the final library prep stages. The two libraries were pooled and sequenced on the NovaSeq 6000 (Illumina) SP 150 bp paired-end format (The Ramaciotti Centre for Genomics).

HiFi reads were generated using the circular consensus sequencing algorithm in SMRT Link v9.0.0.92188 and assembled using PacBio's Improved Phased Assembler v1.1.2 ('URLs' section). The Purge_dups v1.2.3[56] was used to remove haplotigs and contig overlaps from both the primary and alternative assemblies. An interleaved linked reads file was created from the raw 10x Genomics reads using Long Ranger v2.2.2[57] and aligned to the draft assembly with Burrows–Wheeler Aligner mem v0.7.17-r1188[58]. The output was sorted using samtools v1.9[59] and scaffolding was performed using ARCS v1.1.1[60] and LINKS v1.8.7[61] with the -D option to estimate gap sizes. PBJelly v15.8.24[62] was used for gap filling the scaffolded assembly with default parameters and Pilon v1.20[63] to polish the final assembly using the 10x reverse reads that were quality trimmed (trimming parameters: ftl=10 trimq=20 qtrim=rl) using BBDuk v37.98[64]. Vector contamination, low quality scaffolds and remaining false duplications were removed using Diploidocus 'dipcycle'[65] with the HiFi reads used for depth analysis and the trimmed 10x paired-end reads used for k-mer analysis. Scaffolds flagged as repeats were put aside and the core genome prepared for HiC scaffolding. Scaffolding based on Omni-C data was carried out with HiRise v2.1.6[66]. The assembly was manually curated by iteratively generating and analysing the Omni-C contact map. Ligation junctions were identified and Omni-C pairs generated using pairtools v0.3.0[67]. Subsequently, we generated a multi-resolution Omni-C matrix in binary form with cooler v0.8.10[68] and balanced it with hicExplorer v3.6[69]. We used HiGlass v2.1.11[70] and the PretextSuite ('URLs' section) to visualize the contact maps. This resulted in the Ninu reference genome v1.9. This comprised a 3.66 Gb genome, including 609 scaffolds, with a scaffold N50 of 343.8 Mbp (Extended Data Table 2). The two largest chromosome scaffolds were too big for some tools, so a version of the genome was also created with each of these scaffolds split into two subscaffolds (Supplementary Fig. 5).

Early assessments of assembly completeness were conducted with BUSCO v3.0.2b[71] (mammalia_odb9; 4,104 genes). Subsequently, BUSCOMP v1.1.2 (ref. [72]) and BBTools v38.73 (ref. [64]) were used to generate general assembly statistics and assembly completeness was assessed using BUSCO v5.4.4 (ref. [71]) (mammalia_odb10; 9,226 genes). The assembly showed high completeness with 0.34% gaps, 92.2% Merqury k-mer completeness and 93.5% complete mammalian BUSCOs (Extended Data Table 2) (5.1% missing). The BUSCO duplication rate remains quite high at 4.9%, possibly as a consequence of the low HiFi sequencing depth reducing the power of depth-based removal of false duplications.

**Resequenced genomes.** A total of 12 Ninu genomes were resequenced, 6 individuals (3 males, 3 females) from a temperate ancestry and 6 individuals (3 males, 3 females) from a semi-arid ancestry. In addition, five Yallara collected between 1895 and 1931 were sampled (Supplementary Table 1 and Supplementary Note 1.7). Ninu DNA was extracted from ear biopsies stored in 70% ethanol using MagAttract HMW DNA Kit (Qiagen catalogue 67563). A TruSeq DNA polymerase chain reaction-free library prep (Illumina) was used, and samples sequenced as 150 bp paired-end reads across a single S2 flowcell on the Illumina NovaSeq 6000 obtaining ~30× coverage per sample. Yallara samples were extracted in a Trace DNA laboratory using a modified protocol from Fulton, Wagner[73] and a Qiagen DNeasy Blood and Tissue kit (Qiagen catalogue 69504). One sample failed QC (NMV C7091) and so the remaining four had ThruPLEX DNA (Takara Bio) library prep and were sequenced on an Illumina NovaSeq 6000 S1 as 2 × 150 bp paired-end reads. Two of the best coverage samples (Supplementary Table 1) underwent a Meyer Kircher library prep[74] and Illumina adaptor ligation with a dual 8 bp index to obtain higher coverage (Supplementary Table 1).

The resequenced genomes from both species were aligned to v1.5 of the reference genome and variants called using the DRAGEN Germline platform v3.8.4 (Illumina)[75]. Joint genotyping across all 12 Ninu samples was also performed with DRAGEN Joint Genotyping v3.8.4. Bcftools v1.11 (refs. [76,77]) was used to split multi-allelic variant calls and to left-normalize the variants before variant annotation with ANNOVAR v20180416 (ref. [78]). Genotyping rates were calculated using PLINK v1.90 (ref. [79]). A GWAS was performed on the 12 Ninu resequenced

genomes to identify allele frequency differences between the temperate and semi-arid samples. For full methods see Supplementary Note 1.8. In summary, the reference genome was indexed with Picard v2.21.9 (ref. 80) and SAMtools v1.6 (refs. 59,76), and the joint genotyping variant call format (VCF) was filtered using the Genome Analysis Toolkit v4.2.0.0 (ref. 81) and VCFtools v0.1.14 (ref. 82) to retain only bi-allelic SNPs, with no missing data and a minor allele frequency >0.05. To mitigate small sample sizes, three association tests were performed as per Batley et al.[83] using either PLINK v1.90 (ref. 79) or VCFtools. BEDtools v2.29.2 (ref. 84) identified genes containing candidate SNPs and unique genes were run through GONet[85] to obtain a network of biological processes with GO term annotation and visualized with Revigo[86].

**Ninu population genomics.** We inferred the historical effective population size of Ninu ($N = 12$) and Yallara ($N = 2$) using MSMC and PSMC models in MSMC2[87]. Five separate analyses were conducted to observe differences across Ninu species and populations (Supplementary Note 2.1). Due to the computational limitations of MSMC, four individuals (eight haplotypes) with the highest mean sequencing coverage were selected for all three Ninu MSMC analyses (Supplementary Table 8). Two Yallara high coverage individuals (four haplotypes) were used (Supplementary Table 8). Twenty bootstrap replicates were run for each of the two species-level MSMC analyses. All analyses used a time interval of -p 1*3+10*1+1*3 to prevent overfitting and scaled using the estimated mutation rate of the Tasmanian devil ($1.17 \times 10^{-9}$ mutations per site per generation)[88].

ROH in the 12 Ninu resequenced genomes were characterized using PLINK v1.9 (ref. 79) (Supplementary Note 2.2). Putatively sex-linked scaffolds and all missing data were removed. After filtering, 29,266,950 SNPs remained. We chose PLINK settings for the ROH analysis of this high-density SNP dataset following recommendations in Kardos et al.[89], Ceballos et al.[90] and Meyermans et al.[91]. A sliding window of 50 SNPs was run using PLINK; homozygous regions of at least 100 kb and 100 SNPs were considered ROH. One heterozygous SNP per window was permitted for genotyping error. A minimum of one SNP per 50 kb was required to call a ROH and the maximum gap allowed between two SNPs was 200 kb. At least 5% of windows were required to contain a given homozygous SNP for it to be considered within a ROH. We reran these ROH analyses applying various parameter combinations to ascertain the sensitivity of the results to the choice of parameter (Supplementary Table 9). $F_{ROH}$ was calculated for all ROH >100 kb ($F_{ROH>100kb}$), 500 kb ($F_{ROH>500kb}$) and 1 Mb ($F_{ROH>1Mb}$) to compare across species. Observed heterozygosity was calculated for each individual based on the same SNP dataset using VCFtools.

Samples were obtained from 13 contemporary metapopulation locations, including zoos, as well as the wild populations (Pilbara, Kimberley[92], Birdsville and Currawinya) (Table 1). A total of 363 Ninu were sampled between 2011 and 2022 (Supplementary Note 2.4). DNA was extracted using the DNeasy blood and tissue kit (Qiagen catalogue 69504) or the MagAttract HMW DNA Kit (Qiagen catalogue 67563). Pilbara biopsies were extracted with a standard salting out extraction protocol[93] with the addition of 3 µl 10 mg ml⁻¹ RNase A (Omega Biotek, catalogue AC118) to the TNES buffer to remove RNA contamination. All extracted DNA samples were sequenced with DArTseq Pty Ltd using a *PstI–SphI* enzyme combination[94] on a HiSeq 2500 (Illumina) as 77-bp, 83-bp or 138-bp single-end reads. Variants were called and filtered using previously published methods[95,96] (Supplementary Note 2.4).

We separated the metapopulation genetic analyses into three groupings (Table 1 and Supplementary Note 2.4). Observed heterozygosity, expected heterozygosity and allelic richness were calculated using the hierfstat package v0.5-10[97] in R. Genetic differentiation was visualized using principal coordinate analysis with the dartR package v1.9.9.1[98]. Inbreeding coefficients ($F_{IS}$) were calculated with the diveRsity package v1.9.90 and 1,000 bootstraps were used to estimate 95% confidence intervals (CIs)[99]; population mean kinship

(MK) was calculated by averaging pairwise comparisons estimated with COANCESTRY v1.0[100]. Pairwise $F_{ST}$ values were calculated with the StAMPP package v1.6.3 and 2,000 bootstraps used to estimate the 95% CIs[101]. NEstimator v2.1[102] was used to estimate effective population size, with a fastSTRUCTURE analysis[103] performed to estimate the number of genetic clusters, $K$, testing $K = 1–10$ clusters with 10,000 iterations for each $K$. The 'chooseK.py' script was used to select the optimum $K$.

Scat samples and metadata were collected by the Kiwirrkurra Indigenous rangers. Kiwirrkurra is located in the 'tali' (sandhill) country of the Gibson Desert (Fig. 3a) and has been described as the most remote community in Australia[104]. The entire 45,867 km² Kiwirrkurra native title determination is managed as an Indigenous Protected Area. Residents speak Pintubi or a mix of other western desert languages. Under the guidance of Traditional Owners, and with assistance from Desert Support Services, the Kiwirrkurra rangers undertake cultural burning, feral animal and weed control, threatened species monitoring, and passing knowledge from elders to young people. Many Kiwirrkurra community members still engage in traditional land-use practices[104]. Full development of the MassARRAY panel is in Supplementary Note 2.5. In brief, we used SNP loci identified from re-mapping raw DArT-seq reads from tissue samples. A total of 35,039 SNPs were identified and filtered with dartR v1.9.6[98] and SNPRelate v0.9.19[105] to obtain high-quality, informative SNP loci for the panel design. Using the male WGR data aligned to the female reference genome, we identified two marsupial Y genes (*KDM5D* and *HCFC1*) that were suitable for sexing scat samples. After preliminary testing with scat and tissue samples, we developed 35 autosomal and four sex-linked markers for scat genotyping. SNP genotyping was carried out on the MassARRAY system (Agena Bioscience). Amplification and extension reactions were performed using the iPLEX Gold Reagent Kit (Agena Bioscience) according to the manufacturer's protocols using 1 µl of tissue or faecal DNA. Resultant SNP genotypes were identified by mass spectrometry and called using MassARRAY TyperAnalyzer 4.1 software (Agena Bioscience) by the Australian Genome Research Facility. We included ~10% repeats to ensure consistency across runs and to calculate the genotyping error rate.

**Genome annotation and gene family analysis.** Total RNA was extracted from 25 mg of each tissue from the reference female Ninu (spleen, liver, lymph node, kidney, heart, tongue, ovary, uterus, pouch skin, mammary gland and salivary gland) and from blood using the RNEasy Protect animal blood kit (Qiagen catalogue 73224). Total RNA was also extracted from testis tissue from a single male Ninu. RNA was quantified on a Bioanalyzer RNA 6000 Nano Kit (Agilent Technologies catalogue 5067-1511) before TruSeq stranded total RNA library preparation (Illumina), with ribosomal RNA depletion using the Illumina Ribo-zero gold kit. A total of 12 tissue libraries from the reference female Ninu were sequenced on a S1 flowcell with 150 bp paired-end reads on the Illumina NovaSeq 6000 at the Ramaciotti Centre for Genomics (University of New South Wales). The testis library was sequenced on a S1 flowcell with 100 bp paired-end reads on the Illumina NovaSeq 6000 at Ramaciotti Centre for Genomics. Raw RNA sequencing reads (~100 million reads per sample) underwent quality and length trimming using Trimmomatic v0.38[106] in paired-end mode. For the global transcriptome of 12 tissues, trimmed reads were aligned to the genome v1.5 using HISAT2 v2.1.0[107] with default parameters and alignments were converted and sorted using samtools. Transcripts were assembled using StringTie v2.1.3 (ref. 108) and the resulting transcript models across the tissues were merged into a single global transcriptome using TAMA merge v0.0 (ref. 109). Transcriptome completeness was assessed using BUSCO v5.4.6 as above. TransDecoder v2.0.1 (ref. 110) was used to determine coding regions and open reading frames within transcripts. Following genome annotation, transcripts were assembled using StringTie v2.1.3 with the GeMoMa genome annotation as a guide (Supplementary Note 3.3) to generate fragments per kilobase of

transcript per million mapped reads counts for each transcript within the global transcriptome. The testis transcriptome was generated using the same workflow as above.

The global transcriptome was generated as above and aligned to version 1.5. It was composed of 39,106 genes and 303,420 isoforms (including non-coding transcripts) with an average transcript length of 6,833 bp and an N50 of 13.4 kb (Extended Data Table 3). For all protein-coding transcripts, the longest open reading frame had an average transcript length of 1,010 bp and N50 of 1,620 bp. A homology-based annotation was created using GeMoMa v1.8 (ref. 111) using the annotation from ten mammalian genomes (cow, human, opossum, mouse, Tammar wallaby, platypus, koala, Tasmanian devil, wombat and brown antechinus) (Supplementary Table 16). GeMoMa annotated 63,480 isoforms for 38,756 genes, with a median Ninu:opossum protein length ratio of 0.986 versus the *Monodelphis domestica* reference proteome. This was similar to the 39,106 genes in the global transcriptome and was rated as 96.0% complete by BUSCO v5 (proteome mode) (Extended Data Table 3). The average GeMoMa gene prediction was 1,120 bp (lacking untranslated regions) with an average of 6.32 exons per gene. RepeatModeler v2.0.1 (ref. 112) was used to create a custom repeat database using the HiC-scaffolded genome. In total, 47.87% of the assembly was annotated as interspersed repeats, with L1 LINEs being the dominant repeat type (20.91% assembly), and a further 6.22% as low complexity and simple repeats. Synteny plots were created with GENESPACE v1.3.1 (ref. 113) against five other Australidelphia marsupials (Fig. 4), in addition to the Ameridelphia and *Homo sapiens* (Supplementary Fig. 4).

To investigate the evolution of gene family size and avoid inflated estimates of gene family differences, we compared protein sequences re-annotated using GeMoMa v1.8 (ref. 111) for other marsupials (opossum, Tasmanian devil, koala, brown antechinus and Tammar wallaby), eutherians (human, mouse and cow) and a monotreme (platypus) (Supplementary Note 3). To control for pseudogenes, we removed genes annotated as 'Predicted protein' or 'Reverse transcriptase homologues' from further analyses. Orthologous genes were identified with OrthoFinder v2.4.01 using default settings. A dated species tree was constructed using MCMCTree in PAML v4.9 (ref. 114) following Jeffares et al.[115]. We tested for expansions of gene families under a birth-death model using CAFE v.5.0 (ref. 116). The gene counts from OrthoFinder and dated species tree from MCMCTree were used as inputs for CAFE. To minimize the impact of gene families (orthogroups) with highly variable gene counts when estimating lambda, gene families with 100 or more genes in any one lineage were analysed separately. Across all ten species, 74,591 genes were annotated as 'predicted protein' and 123,379 genes were annotated as 'reverse transcriptase homologue'. These were omitted from further analyses, leaving a total of 197,970 annotated genes (Supplementary Table 18).

Olfactory receptor genes were analysed by using the raw gene counts from GeMoMa. *OR1D2* and *OR1D5* were investigated for expansions. The GWAS identified four genes involved in spermatogenesis (*SPEF2*, *TBC1D21*, *SYNE1* and *NME8*). Protein sequences of these were extracted from the reference assembly and BLASTp v2.2.30 (refs. 117,118) was used to determine expression in the testis. The protein sequences of a set of 115 'core placenta' genes (Supplementary Excel) were extracted from Ensembl genomes v104[119] of *Monodelphis domestica, Rattus norvegicus*, and *Mus musculus*. BLASTp v2.2.30[117,118] was used to determine presence/absence of these 'core placenta' genes in the Ninu uterus transcriptome assembly.

Immune genes were annotated using multiple search strategies depending on the type of gene family (Supplementary Note 3.8). In general, a combination of BLAST v2.2.30[117], hidden Markov models constructed using Clustal-W alignments and HMMER v3.2 (ref. 120) were used to search the Ninu reference assembly, associated annotation files and/or transcriptomes using published marsupial, monotreme and eutherian immune gene sequences as queries. Putative gene sequences were queried against the Swiss Prot nonredundant database, and any sequences with top hits to Swiss Prot genes, marsupial-specific genes or other domain models were retained (Supplementary Note 3.8). Putative immune genes were named following the appropriate nomenclature for each family, with duplicated genes named according to their genomic location from the 5′ to 3′ end of the locus. MHC Class I and II genes were named on the basis of their evolutionary relationship with other marsupial MHC genes. Phylogenetic trees were constructed using the neighbour-joining method[121] with 1,000 bootstrap replicates[122] in MEGA11 (ref. 123). Genes with clear homologous relationships to marsupial MHC genes were assigned names on the basis of their marsupial counterparts. Genes with no clear relationship were assigned species-specific names.

### URLs
See the URL links for further information on the PacBio Improved Phase Assembler (https://github.com/PacificBiosciences/pbipa) and Pretext-Suite (https://github.com/wtsi-hpag/PretextMap; https://github.com/wtsi-hpag/PretextView; https://github.com/wtsi-hpag/PretextSnapshot).

### Ethics and inclusion statement
Our large collaborative project aimed to use genomic technologies to develop new management tools for the conservation of the Ninu; as a result, our authorship includes early, mid and late career academic researchers from Australian and international universities, Indigenous Australians and Australian conservation managers.

Tissue samples for the reference individual, and the male testis, were collected opportunistically when individuals were euthanized for medical purposes. Ear biopsies are collected as part of the metapopulation routine monitoring programmes, or during targeted trapping and capture events, that they conducted in accordance with the standard operating procedures for each organization. These management samples were shared with us as part of a study plan approved by representatives from the participating ZAA facilities, the AWC, the Australian Museum, the University of Sydney and the Greater Bilby National Recovery Team Metapopulation Committee.

### Reporting summary
Further information on research design is available in the Nature Portfolio Reporting Summary linked to this article.

## Data availability
Raw and processed data for the reference genome, transcriptomes and resequenced genomes are available via NCBI for the Ninu (PRJNA1049866) and Yallara (PRJNA1049868), in addition to via the Australasian Genomes website at https://awgg-lab.github.io/australasiangenomes/genomes.html. The DArTseq SNP genotypes for population genetic analysis and the MassARRAY scat genotyping assay are available via Dryad at https://doi.org/10.5061/dryad.gtht76htz.

## Code availability
The code used to select SNPs to design the custom MassARRAY scat genotyping assay is provided via Dryad at https://doi.org/10.5061/dryad.gtht76htz. All other analyses used standard software and scripts as described in Methods and Supplementary Information.

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

## Acknowledgements

We acknowledge the traditional custodians of the land upon which bilbies have been, and are still, found and pay respects to their elders past and present. There are a massive number of individuals to acknowledge as part of this project. We thank Perth Zoo and Taronga Zoo for the provision of the individuals (female and male, respectively) for the reference genome and transcriptomes. Thanks also to A. Sayer for assistance in acquiring the genome sample from Perth Zoo. We thank staff and teams at the Australian Wildlife Conservancy, Arid Recovery, Department for Environment and Water (South Australia), Save the Bilby Fund and the Zoo and Aquarium Association (ZAA) member institutions holding bilbies for their assistance with sample collection from the bilby metapopulation. We also thank the Kiwirrkurra Indigenous rangers and high school students for their collection of scat samples. The lesser bilby samples were provided by the Natural History Museum, London (NHMUK), South Australian Museum (SAMA) and Museum Victoria (NMV). Technical support for the contemporary tissue extractions were provided by K. Heasman and L. Alexander (University of Sydney) and for the original lesser bilby extractions by A. MacDonald (ANU), the scat samples by S. McArthur and M. Millar (DBCA) and T. Bertozzi for the Illumina library prep kits for the lesser bilby samples. We also thank M. Dziminski and F. Carpenter (DBCA) for their collection and extraction of the Pilbara tissue samples. The authors acknowledge the technical support provided by the Sydney

Informatics Hub, a Core Research Facility of the University of Sydney, particularly T. Chew, C. Willet and R. Sadsad for the transcriptome pipeline, the earlier WGS NCI Gadi pipeline and the 10x assembly. The WGR alignment and variant calling was performed on the Illumina DRAGEN pipeline, we thank the DRAGEN development team for their assistance. Cloud support was provided by Amazon Web Services, Queensland's Research Computing Centre (RCC) FlashLite facility, the Pawsey Supercomputing Centre for genome assembly and annotation, and RRS data analysis. We thank the Vertebrate Genomes Project (VGP) members for early pre-publication access to several genomes, in particular G. Myers, E. Jarvis, R. Nespolo and colleagues, and for *Trichosurus vulpecula* from E. Jarvis, N. Gemmell, T. Hore, M. Laird and colleagues. The 10x Genomics sequencing was provided as part of the Oz Mammals Genomics (OMG), and bioinformatic support was provided as part of the Threatened Species Initiative (TSI), both supported by funding from the Australian Government National Collaborative Research Infrastructure Strategy (NCRIS) Bioplatforms Australia. The PacBio HiFi sequencing and HiC was supported by the University of Sydney through research grants to K.B. (DP180102465) and C.J.H. (Toledo Zoo and Aquarium). K.F., E.P., C.J.H. and K.B. (CE200100012), C.M.W. (DP180103370), and F.R.J. and S.Y.W.H. (FT160100167) received funding from the Australian Research Council; K.T. was supported by an Australian Biological Resources Study (ABRS) grant, and M.R.W. and the Ramaciotti Centre for Genomics acknowledge funding from the Australian Government NCRIS programme and Bioplatforms Australia. A.R.H. received funding from the Spanish Ministry of Science and Innovation (PID2020–112557GB-I00) and the Agència de Gestió d'Ajuts Universitaris i de Recerca, AGAUR (2021 SGR 122). L.M.-G. was supported by the Ministry of Science, Innovation and University (FPU18/03867 and EST22/00661). P.D.W. is supported by the Australian Research Council (DP170101147, DP180100931, DP210103512 and DP220101429) and NHMRC Ideas Grant (2021172). R.J.E. was supported by the Australian Research Council (LP18010072).

## Author Contributions

C.J.H. coordinated the project, undertook the metapopulation analyses and recommendations, and compiled the figures. R.J.E. assembled and curated the final genome assembly. K.A.F. undertook the metapopulation analyses, aligned resequenced genomes and undertook comparative analyses. L.W.S., P.B., E.P., M.E., F.R.J., R.T., K.B., T.M.B., J.K.C., Z.C., N.D., M.D., K.M.E., O.W.G., L.M.G., K.L.M., K.J.T., P.W., C.M.W., M.R.W., K.M.H., N.L., S.Y.W.H., A.R.H., R.P., J.A.M.G., M.R., B.S. and K.O. performed additional analyses and contributed to intellectual discussions. C.J.H. and K.B. sourced funding and wrote the manuscript.

## Funding

## Competing interests

The authors declare no competing interests.

## Additional information

**Extended data** is available for this paper at https://doi.org/10.1038/s41559-024-02436-2.

**Correspondence and requests for materials** should be addressed to Carolyn J. Hogg.

Carolyn J. Hogg ●[1,2,21]✉, Richard J. Edwards ●[3,4,21], Katherine A. Farquharson ●[1,2,21], Luke W. Silver[1], Parice Brandies[1], Emma Peel[1,2], Merly Escalona ●[5], Frederick R. Jaya[1], Rujiporn Thavornkanlapachai[6], Kimberley Batley[1], Tessa M. Bradford ●[7,8], J. King Chang ●[4], Zhiliang Chen[9], Nandan Deshpande[4,10], Martin Dziminski[6], Kyle M. Ewart[1], Oliver W. Griffith ●[11], Laia Marin Gual ●[12,13], Katherine L. Moon[5,14], Kenny J. Travouillon[15], Paul Waters ●[4], Camilla M. Whittington ●[1], Marc R. Wilkins[4,10], Kristofer M. Helgen ●[16], Nathan Lo ●[1], Simon Y. W. Ho ●[1], Aurora Ruiz Herrera ●[12,13], Rachel Paltridge[17], Jennifer A. Marshall Graves[18], Marilyn Renfree ●[19], Beth Shapiro[5,14], Kym Ottewell ●[6], Kiwirrkurra Rangers* & Katherine Belov[1,2]

[1]School of Life and Environmental Sciences, The University of Sydney, Sydney, New South Wales, Australia. [2]ARC Centre of Excellence for Innovations in Peptide and Protein Science, The University of Sydney, Sydney, New South Wales, Australia. [3]Minderoo OceanOmics Centre at UWA, Oceans Institute, The University of Western Australia, Perth, Western Australia, Australia. [4]School of Biotechnology and Biomolecular Sciences, UNSW Sydney, Sydney, New South Wales, Australia. [5]Department of Ecology and Evolutionary Biology, University of California Santa Cruz, Santa Cruz, CA, USA. [6]Biodiversity and Conservation Science, Department of Biodiversity, Conservation and Attractions, Kensington, Western Australia, Australia. [7]Evolutionary Biology Unit, South Australian Museum, Adelaide, South Australia, Australia. [8]School of Biological Sciences, The University of Adelaide, Adelaide, South Australia, Australia. [9]Illumina, Melbourne, Victoria, Australia. [10]Ramaciotti Centre for Genomics and School of Biotechnology and Biomolecular Science, UNSW, Sydney, New South Wales, Australia. [11]School of Natural Sciences, Macquarie University, Sydney, New South Wales, Australia. [12]Departament de Biologia Cel·lular, Fisiologia i Immunologia, Universitat Autònoma de Barcelona, Cerdanyola del Vallès, Spain.

# Article

[13]Genome Integrity and Instability Group, Institut de Biotecnologia i Biomedicina, Universitat Autònoma de Barcelona, Cerdanyola del Vallès, Spain. [14]Howard Hughes Medical Institute, University of California Santa Cruz, Santa Cruz, CA, USA. [15]Collections and Research, Western Australian Museum, Welshpool, Western Australia, Australia. [16]Australian Museum Research Institute, Australian Museum, Sydney, New South Wales, Australia. [17]Indigenous Desert Alliance, Alice Springs, Northern Territory, Australia. [18]Department of Environment and Genetics, La Trobe University, Melbourne, Victoria, Australia. [19]School of BioSciences, University of Melbourne, Melbourne, Victoria, Australia. [21]These authors contributed equally: Carolyn J. Hogg, Richard J. Edwards, Katherine A. Farquharson. *A list of authors and their affiliations appears at the end of the paper. ✉e-mail: carolyn.hogg@sydney.edu.au

## Kiwirrkurra Rangers

**Conway Gibson**[20]**, Raymond Maxwell**[20]**, Zecharia Spencer**[20]**, Yalti Napangati**[20]**, Mary Butler**[20]**, Janine West**[20]**, John West**[20]**, Mantua James**[20]**, Nolia Napangati**[20]**, Loretta Gibson**[20]**, Payu West**[20]**, Angus Gibson**[20]**, Scott West**[20]**, Kim West**[20]**, Walimpirri Japaltjari**[20]**, Ed Blackwood**[20] **& Rachel Paltridge**[20]

[20]Kiwirrkura Community, Gibson Desert, Western Australia, Australia.

**Extended Data Table 1 | Indigenous names for Greater bilbies**

| Language group (and area) | Bilby and related names |
|---|---|
| Pintupi (Great Sandy Desert) | Ninu, Natari, Taaalku, Kuninka |
| Pintupi (Great Sandy Desert) | Patiri ~ bandicoot's tail |
| Warlpiri (Tanami desert) | Walpajirri |
| Warlpiri, eastern | Pingki-tawutawu |
| Manjilyjarra, Martu wangka | Mankarr |
| Martu dialect | Kulkawalu |
| Martu at Birriliburu | Muntalngaku * |
| Ngaanyatjarra (Warburton region) | Marrura, Ninu |
| Warlmanpa (NNW Tennant Creek) | Warrikirti |
| Anmatyerr (Ti Tree region) | Angkay |
| Pitjantjatjara (South Australia) | Ninu |
| Yankunytjarra | Tjalku |
| Walmajarri | Nyalku/Nyarlku/Nyarlgoo/Kurrmili |
| Walmajarri (Ngurrara) | Mirtuluju |
| Ullaroi/Yuwaalayaay (NW NSW) | Bilba |
| Kimberley language names | Nyarlku/ Nyarlgu, Nyarlgoo, Nyarlku, Nalgo, Nalgo-midi, Jitarru/ Jidardu, Kurmili, Kurrmili, Mirtulurtu, Birndirdiri, Yawuri, Gurmin |
| To be determined | Yinpu |
| To be determined | Nirlyari |
| To be determined | Jawinji |
| To be determined | Jirrartu |
| Kimberley language | Birndirdiri |
| **Bilby and bilby tail names from other Central Australian languages** | |
| Alyawarr (Sandover) | Ahert |
| Arrernte (Alice Springs region) Western, Central, Western C. Strehlow | Ahert, Kere Aherte Kere angkaye Ngkaye, Kere ngkaye Inkaia Albitja |
| Arrernte (Alice Springs region) | Alpirte, Alpitye |
| Kayteye (Barrow Creek region) | Artnangke |
| Kayteye (Barrow Creek region) | Alpite |
| Jingulu & Mudburra (Elliott region) | Yarningki |
| Kuwarri (Elliott region) | Yalbawurrini |
| Alekerange area | Jawinji, Walpajirri, Nini, Ngarlaparaji, Pingki-tawu-tawu (all the same animal) |
| Nyungar/ Nyoongar (SW WA) | Dalgayt/ Dalgyte/Dalgite/Dol-goitch |
| Antikirinya (Port Augusta) | Malku (Bilby or possibly Bilby-tail) |
| Ooldea | Milbu (Bilby tail) |

Greater bilbies are culturally important to many Indigenous Australians and go by many names. Listed below are some of the names given to the Greater bilby by mobs across Australia, modified from the report produced as part of the Ninu Festival at Kiwirrkurra in 2016[1]. This is by no means a comprehensive list but rather represents the many Indigenous communities of which this species has cultural importance and value, including giving a separate name to the bilby tail in some languages.

**Extended Data Table 2 | Comparative genome statistics**

| Species | Ninu (Greater bilby, *Macrotis lagotis*) This study | Yallara (Lesser bilby, *Macrotis leucura*) This study | Tasmanian devil (Purinina, *Sarcophilus harrisii*)[127] | Koala (Guba, *Phascolarctos cinereus*)[17] | Woylie (Brush-tailed bettong; *Bettongia penicillata ogilbyi*)[128] | Short-tailed opossum (*Monodelphis domestica*)[129] |
|---|---|---|---|---|---|---|
| Genome assembly version | v1.9 | v1.0 | mSarHar1.11 | phaCin_unsw_v4.1 | mBetpen1.pri.20210916 | mMonDom1.pri |
| Data type | PacBio HiFi, Dovetail HiC, Illumina, 10x | Illumina | ONT 10x ULR, 10x Chromium, BioNano, Dovetail HiC | PacBio RSII, Illumina, BioNano | PacBio HiFi, Illumina | PacBio HiFi, BioNano, Arima HiC |
| Genome size (Gbp) | 3.66 | 3.50 | 3.10 | 3.19 | 3.39 | 3.59 |
| GC (%) | 37.35 | 36.12 | 36.16 | 39.05 | 38.64 | 37.92 |
| No. scaffolds | 609 | 10 | 106 | 1,909 | 1,116 | 14 |
| Scaffold N50 (Mbp) | 343.85 | 344.21 | 611.35 | 11.59 | 6.94 | 538.30 |
| Chromosome 1 (bp) | 934,426,298 | NA | 716,413,629 | N/A | N/A | 760,810,273 |
| No. contigs | 5,028 | 6,329,012 | 445 | 1,909 | 3,016 | 2,269 |
| Contig N50 (Mbp) | 1.22 | 851 (bp) | 62.34 | 11.58 | 2.00 | 3.91 |
| Complete mammalian BUSCO (v5.3.2) | 93.5% | 75.2% | 91.8% | 94.2% | 94.3% | 92.0% |
| % Gaps | 0.34 | 19.74 | 0.002 | 0.00 | 0.40 | 0.89 |

Genome statistics of the current Ninu genome compared with recently published marsupial genomes.

**Extended Data Table 3 | Genome statistics**

| Statistic | Value |
|---|---|
| **Genome** | |
| Assembly Size | 3.66 Gb |
| Coverage | 39x |
| Assigned to chromosomes | 3.50 Gb (95.6%) |
| No. Chromosomes | 9 nuclear, plus mtDNA |
| No. Scaffolds | 609 (10 with scaffolding) |
| No. Contigs | 5,028 (4,429 with scaffolding) |
| Scaffold N50 | 343.85 Mb |
| Contig N50 | 1.23 Mb |
| Gaps | 0.34% |
| Repeat Content* | 47.45% |
| BUSCO V5 | C:93.5% [S:88.6%, D:4.9%], F:1.1%, M:5.4%, n:9226 |
| No. Protein-coding Genes | 38,756 |
| BUSCO V5 (proteins) | C:96.0% [S:52.4%, D:43.6%], F:0.8%, M:3.2%, n:9226 |
| **Global Transcriptome** | |
| No. Transcripts | 303,420 |
| No. Genes | 39,106 |
| Average Transcript Length | 6,833 bp |
| Transcript N50 | 13.4 kb |
| BUSCO V5 | C:86.1% [S:8.4%, D:77.7%], F:3.0%, M:10.9%, n:9226 |
| **Testis Transcriptome** | |
| No. Transcripts | 82,964 |
| No. Genes | 37,034 |
| Average Transcript Length | 2363 bp |
| Transcript N50 | 3.93 kb |
| BUSCO V5 | C:75.5% [S:25.9%, D: 49.6%], F:4.0%, M:20.5%, n:9226 |

*Excluding low complexity and simple repeats Ninu reference genome and transcriptome statistics.

**Extended Data Table 4 | Comparison of scat and tissue genotyping data**

| Population | Sample | Type | N | $H_O$ | $H_E$ | $F_{IS}$ |
|---|---|---|---|---|---|---|
| Pilbara | Tissue | Wild | 4.7 | 0.291 | 0.428 | 0.229 |
| Pilbara | Scat | Wild | 6.7 | 0.289 | 0.419 | 0.275 |
| Kimberley | Tissue | Wild | 2.9 | 0.371 | 0.387 | −0.058 |
| Kimberley | Scat | Wild | 4.4 | 0.253 | 0.286 | 0.080 |
| Kiwirrkurra (2021) | Scat | Wild | 11.9 | 0.342 | 0.311 | −0.086 |
| Kiwirrkurra (2022) | Scat | Wild | 16.5 | 0.360 | 0.357 | −0.027 |
| Combined 2021-2022 | Scat | Wild | 23.6 | 0.338 | 0.337 | −0.011 |
| Arid Recovery | Tissue | Translocated | 15.5 | 0.390 | 0.396 | 0.040 |
| Currawinya | Tissue | Translocated | 33.7 | 0.429 | 0.436 | 0.015 |
| Mallee Cliffs | Tissue | Translocated | 48.8 | 0.393 | 0.461 | 0.144 |
| Mt Gibson founder | Tissue | Translocated | 15.7 | 0.429 | 0.403 | −0.056 |
| Mt Gibson | Tissue | Translocated | 23.3 | 0.406 | 0.449 | 0.090 |
| Mt Gibson | Scat | Translocated | 6.7 | 0.388 | 0.396 | 0.023 |
| Pilliga | Tissue | Translocated | 36.0 | 0.409 | 0.446 | 0.085 |
| Scotia | Tissue | Translocated | 7.3 | 0.330 | 0.340 | 0.019 |
| Thistle Island | Tissue | Translocated | 28.9 | 0.430 | 0.445 | 0.048 |
| Venus Bay | Tissue | Translocated | 5.0 | 0.326 | 0.304 | −0.073 |
| Yooka1 | Tissue | Translocated | 15.6 | 0.310 | 0.261 | −0.147 |
| Yooka2 | Tissue | Translocated | 3.0 | 0.295 | 0.329 | 0.040 |
| ZAA | Tissue | Zoo | 43.2 | 0.443 | 0.450 | 0.009 |

Average number of individuals genotyped per locus (N), estimates of observed ($H_O$) and expected ($H_E$) heterozygosity and population-level inbreeding ($F_{IS}$) from genotyped scat samples (Kiwirrkurra) compared to wild and translocated Ninu populations across Australia. Data from these latter populations are from the larger DArTseq dataset subset to the 35 autosomal SNPs included in the MassARRAY SNP panel

**Extended Data Table 5 | Number of OR1D2, OR1D5, and total olfactory genes across the ten different species used in the Computational Analysis of gene Family Evolution (CAFE) analysis**

| Species | OR1D2 | OR1D5 | Total |
|---|---|---|---|
| Platypus | 257 | 106 | 363 |
| Cow | 868 | 300 | 1168 |
| Mouse | 733 | 317 | 1050 |
| Human | 401 | 183 | 584 |
| Short-tailed opossum | 847 | 321 | 1168 |
| Tammar wallaby | 629 | 219 | 848 |
| Koala | 648 | 247 | 895 |
| *Ninu* | *980* | *397* | *1377* |
| Tasmanian devil | 809 | 299 | 1108 |
| Brown antechinus | 790 | 302 | 1092 |

Species are in the same order as the phylogenetic tree represented in Extended Data Fig. 3.

**Extended Data Table 6 | Comparison of the immune gene repertoire of the Ninu with two marsupial species with partially invasive placentae (Tasmanian devil and opossum) and non-invasive placentae (koala and woylie)**

| Immune Gene Family | Ninu (Greater bilby) | Tasmanian devil (Purinina)[127] | Short-tailed opossum[129] | Koala (Guba)[17] | Woylie[128] |
|---|---|---|---|---|---|
| Cytokines | 84 | 72 | 76 | 82 | 77 |
| TLR | 10 | 10 | 10 | 10 | 10 |
| MHC-I | 6 | 6 | 6 | 19 | 17 |
| MHC-II | 12 | 8 | 12 | 16 | 23 |
| MHC-III | 38 | 36 | 33 | 39 | 37 |
| Ext. MHC & framework genes | 29 | 32 | 28 | 27 | 31 |
| NKC | 12 | 16 | 15 | 17 | 17 |
| LRC (IG domains) | 92 | 92 | 123 | 51 | 60 |
| Extended LRC | 22 | 16 | 9 | 6 | 22 |
| IG constant | 14 | 11 | 13 | 15 | 20 |
| IG variable | 116 | 61 | 89 | 289 | 226 |
| TCR constant | 9 | 13 | 14 | 10 | 12 |
| TCR variable | 118 | 82 | 67 | 103 | 122 |
| Total | 562 | 385 | 495 | 658 | 674 |

Only species with chromosome length genome assemblies were selected because genome quality influences the ability to characterise immune genes[130].

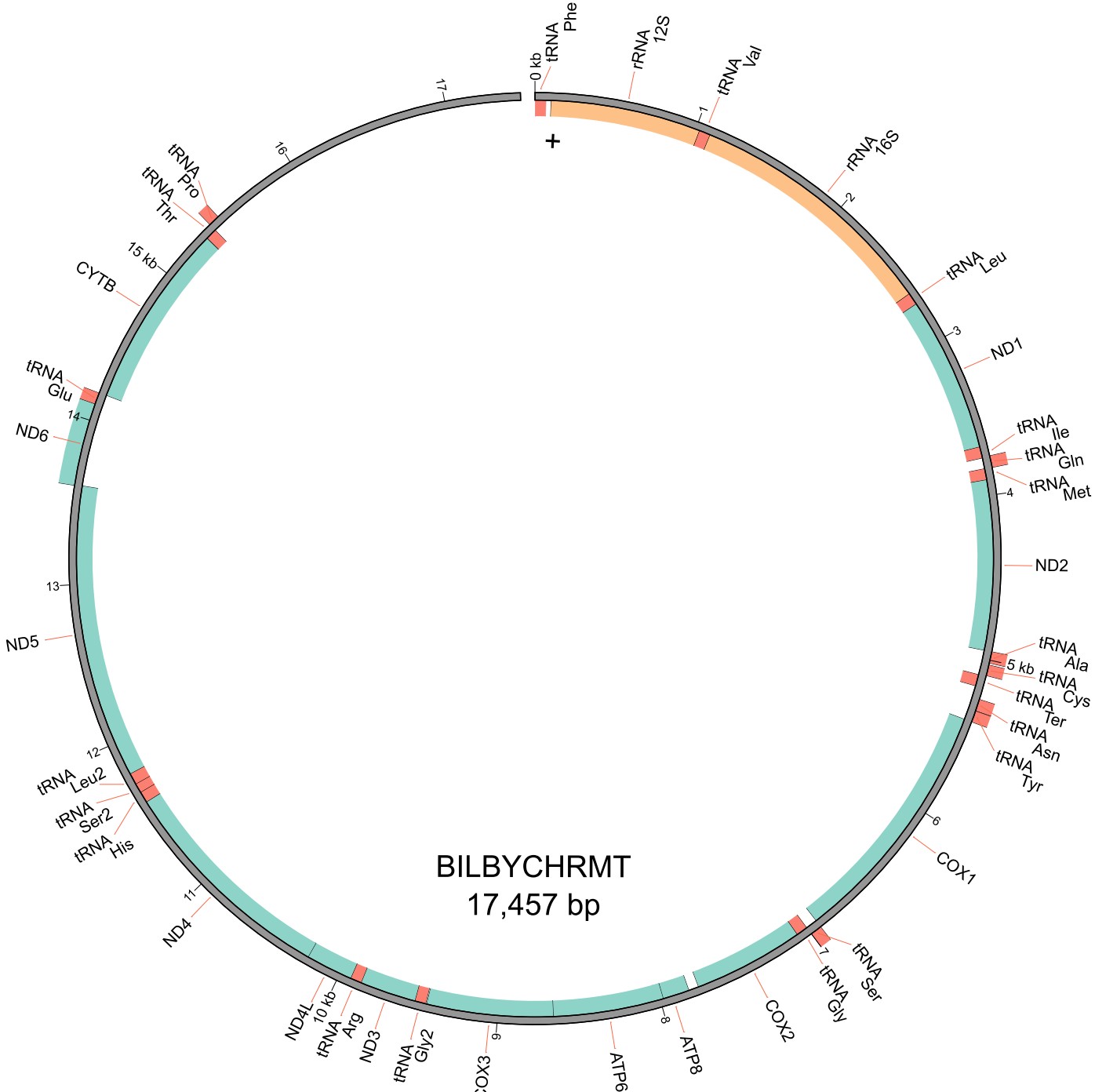

**Extended Data Fig. 1 | Mitochondrial genome.** Complete mitochondrial genome for the Ninu.

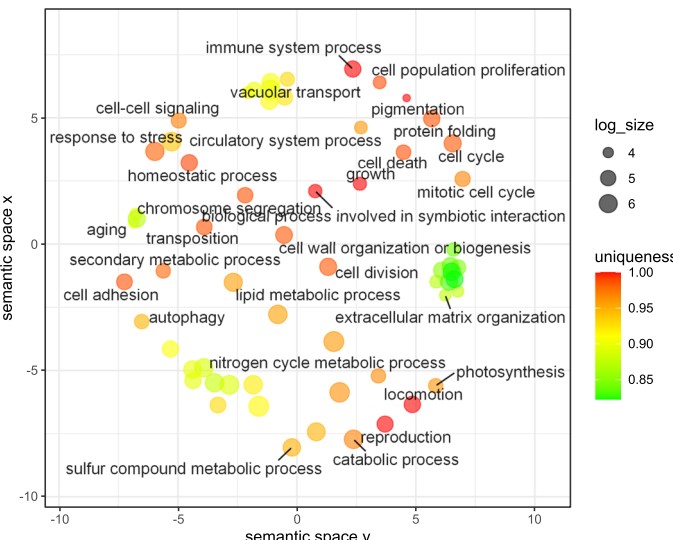

**Extended Data Fig. 2 | Cluster plot of enriched GO terms from association analysis of semi-arid vs. temperate Ninu after semantic reduction with Revigo.** Terms with a dispensability (measure of semantic redundancy) value lower than the median (0.074) are labelled. A full list of enriched GO terms is provided in Table S4. Uniqueness is calculated as 1 minus the average semantic similarity of a term to all other terms.

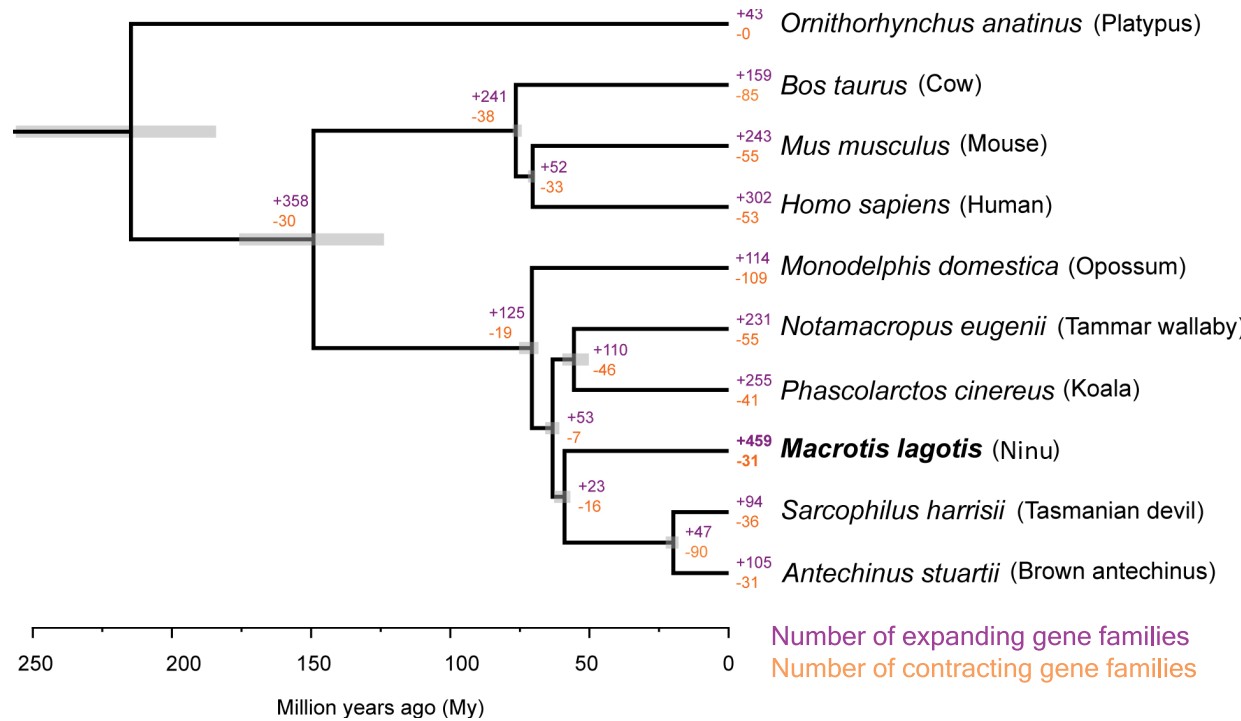

**Extended Data Fig. 3 | Number of gene families significantly expanding and contracting across lineages.** Determined by the Computational Analysis of gene Family Evolution (CAFE) analysis, shown on a dated phylogeny. Grey bars are 95% credibility intervals of divergence-time estimates.

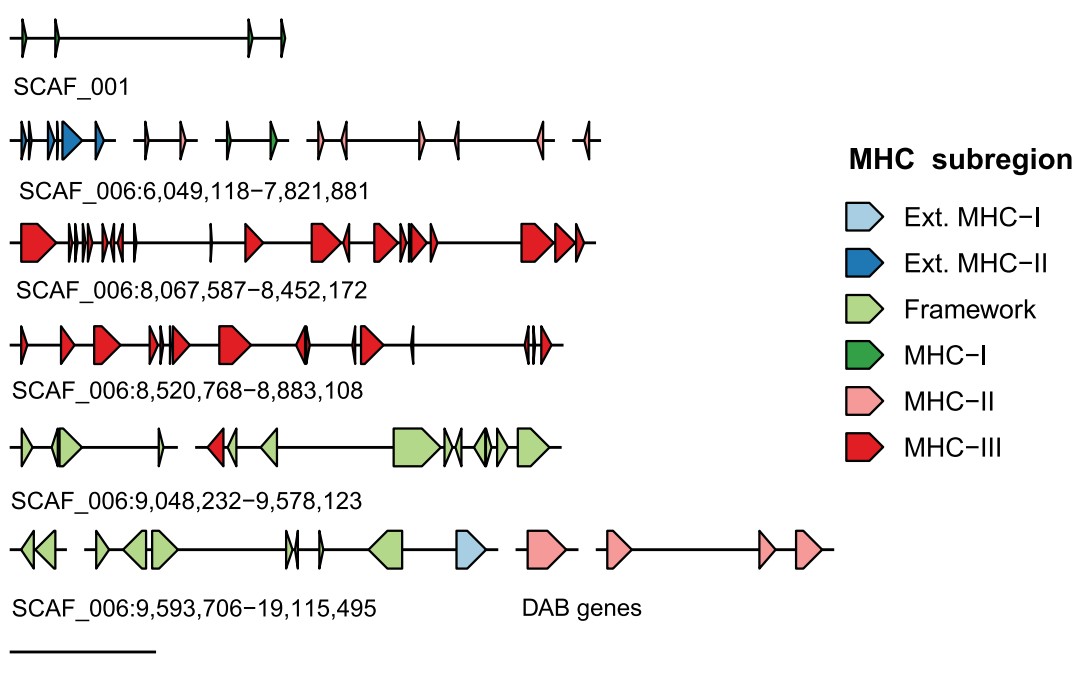

**Extended Data Fig. 4 | MHC region.** Organisation of the Ninu MHC region.

# Reporting Summary

## Statistics

For all statistical analyses, confirm that the following items are present in the figure legend, table legend, main text, or Methods section.

| n/a | Confirmed | |
|---|---|---|
| ☐ | ☒ | The exact sample size (*n*) for each experimental group/condition, given as a discrete number and unit of measurement |
| ☐ | ☒ | A statement on whether measurements were taken from distinct samples or whether the same sample was measured repeatedly |
| ☒ | ☐ | The statistical test(s) used AND whether they are one- or two-sided<br>*Only common tests should be described solely by name; describe more complex techniques in the Methods section.* |
| ☒ | ☐ | A description of all covariates tested |
| ☐ | ☒ | A description of any assumptions or corrections, such as tests of normality and adjustment for multiple comparisons |
| ☐ | ☒ | A full description of the statistical parameters including central tendency (e.g. means) or other basic estimates (e.g. regression coefficient) AND variation (e.g. standard deviation) or associated estimates of uncertainty (e.g. confidence intervals) |
| ☒ | ☐ | For null hypothesis testing, the test statistic (e.g. *F*, *t*, *r*) with confidence intervals, effect sizes, degrees of freedom and *P* value noted<br>*Give P values as exact values whenever suitable.* |
| ☒ | ☐ | For Bayesian analysis, information on the choice of priors and Markov chain Monte Carlo settings |
| ☒ | ☐ | For hierarchical and complex designs, identification of the appropriate level for tests and full reporting of outcomes |
| ☒ | ☐ | Estimates of effect sizes (e.g. Cohen's *d*, Pearson's *r*), indicating how they were calculated |

*Our web collection on statistics for biologists contains articles on many of the points above.*

## Software and code

Policy information about availability of computer code

| Data collection | PacBIO SMRT Link v9.0.0.92188; DRAGEN Germline platform v3.8.4; MassARRAY TyperAnalyzer v4.1 |
|---|---|
| Data analysis | PacBio's Improved Phased Assembler (IPA) v1.1.2; Purge_dups v1.2.3; Long Ranger v2.2.2; Burrows–Wheeler Aligner (BWA) mem v0.7.17-r1188; samtools v1.9; ARCS v1.1.1; LINKS v1.8.7; PBJelly v15.8.24; Pilon v1.20; BBDuk v37.98; HiRise v2.1.6; pairtools v0.3.0; cooler v0.8.10; hicExplorer v3.6; HiGlass v2.1.11; BBTools v38.73; DRAGEN Joint Genotyping v3.8.4. Bcftools v1.11; ANNOVAR v20180416; PLINK v1.90; Picard v2.21.925; GATK v4.2.0.0; VCFtools v0.1.1427; BEDtools v2.29.2; COANCESTRY v1.0; StAMPP package v1.6.3; NeEstimator v2.1; dartR v1.9.6; SNPRelate v0.9.19; HISAT2 v2.1.0; StringTie v2.1.3; TAMA merge v0.0; TransDecoder v2.0.1; StringTie v2.1.3; GeMoMa v1.8; GENESPACE v1.3.1; OrthoFinder v2.4.01; PAML v4.9; CAFE v.5.0; BLASTp v2.2.30; HMMER v3.2; https://github.com/PacificBiosciences/pbipa; https://github.com/wtsi-hpag/PretextMap; https://github.com/wtsi-hpag/PretextView; https://github.com/wtsi-hpag/PretextSnapshot; Stacks v2.61; Trimommatic v0.39; hierfstat package v0.5-10; diveRsity package v1.9.90 |

For manuscripts utilizing custom algorithms or software that are central to the research but not yet described in published literature, software must be made available to editors and reviewers. We strongly encourage code deposition in a community repository (e.g. GitHub). See the Nature Portfolio guidelines for submitting code & software for further information.

# Data

Policy information about availability of data

All manuscripts must include a data availability statement. This statement should provide the following information, where applicable:

- Accession codes, unique identifiers, or web links for publicly available datasets
- A description of any restrictions on data availability
- For clinical datasets or third party data, please ensure that the statement adheres to our policy

Data Availability
Raw and processed data for the reference genome, transcriptomes and resequenced genomes is available via NCBI for the Ninu (PRJNA1049866) and Yallara (PRJNA1049868); in addition to the Australasian Genomes website (https://awgg-lab.github.io/australasiangenomes/genomes.html). The DArTseq SNP genotypes for population genetic analysis and the MassARRAY scat genotyping assay are available at Dryad (https://doi.org/10.5061/dryad.gtht76htz ).

Not included in paper but note for Editor & reviewers, the Dryad link will not go live until paper is accepted, in the meantime the following temporary link can be used to provide reviewers/editors access: https://datadryad.org/stash/share/Y8xsPa_64Po7w4bHE_0xfZJoC9VLBIDk0JlKppSL6ZA.

Code availability
The code used to select SNPs to design the custom MassARRAY scat genotyping assay is provided as Supplementary Text File. All other analyses used standard software and scripts as described in the Methods and Supplementary.

# Research involving human participants, their data, or biological material

Policy information about studies with human participants or human data. See also policy information about sex, gender (identity/presentation), and sexual orientation and race, ethnicity and racism.

| | |
|---|---|
| Reporting on sex and gender | Not Applicable |
| Reporting on race, ethnicity, or other socially relevant groupings | Not Applicable |
| Population characteristics | Not Applicable |
| Recruitment | Not Applicable |
| Ethics oversight | Not Applicable |

Note that full information on the approval of the study protocol must also be provided in the manuscript.

# Field-specific reporting

Please select the one below that is the best fit for your research. If you are not sure, read the appropriate sections before making your selection.

☐ Life sciences    ☐ Behavioural & social sciences    ☒ Ecological, evolutionary & environmental sciences

For a reference copy of the document with all sections, see nature.com/documents/nr-reporting-summary-flat.pdf

# Ecological, evolutionary & environmental sciences study design

All studies must disclose on these points even when the disclosure is negative.

| | |
|---|---|
| Study description | This study characterises the genomes of the extant greater bilby and the extinct lesser bilby. We use this genome to undertake a genetic assessment of bilbies in the metapopulation, and develop and test a new scat SNP array for use by Indigenous Rangers. |
| Research sample | The Greater bilby reference genome was a female individual euthanised for medical reasons - we collected spleen, liver, lymph node, kidney, heart, tongue, ovary, uterus, pouch skin, mammary gland and salivary gland. We also collected testes from a medically euthanised male. The Lesser bilby samples were skin or bone from museums. The metapopulation samples were ear biopsies from 363 individuals;  46 scat samples were collected from a fenced location and wild sites. |
| Sampling strategy | No sample-size calculation was performed as bilbies are extremely difficult to catch. We asked each location to provide samples for all individuals if less than 20 were housed there, and 20 samples for locations with greater than 20 individuals (this is based on previous work that 20-30 individuals captures at least 95% of diversity within a location). |
| Data collection | Metadata for all samples were provided by the institution holding the individuals, this included date collected, individual ID and location. The scat samples were collected by the Kiwirrkurra Indigenous rangers. |
| Timing and spatial scale | The female reference individual was euthanised in 2018; the male was euthanised in 2021. The metapopulation samples were |

| Timing and spatial scale | collected between 2011 and 2022, dates and samples sizes for each are provided in the supplementary. These samples were collected during surveys and translocation events. The scat samples were collected in 2021 and 2022. |
|---|---|
| Data exclusions | Samples were only excluded if poor sequencing coverage was obtained. |
| Reproducibility | Technical replicates were used for both the metapopulation study and the SNP array so we can understand the reproducibility of the sequencing methods and compare results. This data is presented in the supplementary. |
| Randomization | As this was a genetic diversity survey of the whole metapopulation, and development of a new SNP methodology, no randomization was applied. |
| Blinding | As this was a genetic diversity survey of the whole metapopulation, and development of a new SNP methodology, no blinding was applied. |

Did the study involve field work?  ☒ Yes  ☐ No

## Field work, collection and transport

| Field conditions | Samples were collected across Australia, due to the size of the continent it is hard to describe field conditions at each site. Collectively bilbies are located primarily in semi-arid areas, so very dry and hot. Samples were collected during the winter trapping months, and early in the morning. Scat samples were collected in the winter months, early in the morning. |
|---|---|
| Location | Arid Recovery -30.3777293 136.9255; Kimberley (wild) -17.3492 125.9152; Pilbara (wild) -21.7016 120.2511; Scotia -33.39638 141.31305; Thistle Island -35.0381 136.1805; Venus Bay -38.6763 145.7925; Yookamurra -34.5049 139.475358; ZAA -34.91448 138.60651; ZAA -23.70641 133.832568; ZAA -34.9672892 138.696397; ZAA -28.135129 153.488807; ZAA -27.863336 153.315564; ZAA -32.0203936 116.040308; ZAA -35.0888 139.16183; ZAA -33.839442 151.239365; Currawinya -28.84397 144.49557; Dubbo -32.2818 148.57115; Mallee Cliffs -34.21134 142.624833; Mt Gibson -29.63002 117.23731; Pilliga -30.4966464 148.7266; Kiwirrkurra -22.8161 127.7644 |
| Access & import/export | All samples were collected under permit held by the institutions providing the samples, under their standard operating procedures and shared with us for the purposes of managing their populations. All samples were held at the University of Sydney under NSW Scientific Permit SL101204. |
| Disturbance | Bilbies were trapped according to organisational standard operating procedures for the capture, handling and movement of this species. Efforts are made to minmise disturbance whilst trapping and individuals are either trapped and handled at night, or very early in the morning to minimise stress. |

# Reporting for specific materials, systems and methods

We require information from authors about some types of materials, experimental systems and methods used in many studies. Here, indicate whether each material, system or method listed is relevant to your study. If you are not sure if a list item applies to your research, read the appropriate section before selecting a response.

### Materials & experimental systems

| n/a | Involved in the study |
|---|---|
| ☒ | Antibodies |
| ☒ | Eukaryotic cell lines |
| ☒ | Palaeontology and archaeology |
| ☐ | ☒ Animals and other organisms |
| ☒ | Clinical data |
| ☒ | Dual use research of concern |
| ☒ | Plants |

### Methods

| n/a | Involved in the study |
|---|---|
| ☒ | ChIP-seq |
| ☒ | Flow cytometry |
| ☒ | MRI-based neuroimaging |

## Animals and other research organisms

Policy information about studies involving animals; ARRIVE guidelines recommended for reporting animal research, and Sex and Gender in Research

| Laboratory animals | Not applicable |
|---|---|
| Wild animals | The metapopulation samples came from individuals housed in fenced sanctuaries, zoos, or islands. Bilbies were trapped as per standard operating procedures using a cage trap covered with a hessian sack and baited with oats and peanut butter, or captured with a hoop net. Sampling was undertaken at night or very early in the morning. Bilbies were released back to their trapping location, unless they were being translocated as part of the metapopulation management. Wild bilby samples were collected from roadkill, or |

individuals that were trapped and released. The reference genome samples came from a female and male that were medically euthanised at their zoo location.

| | |
|---|---|
| Reporting on sex | All individuals were sexed by the trapping teams by observing their genitalia. |
| Field-collected samples | No individuals were housed in laboratories. |
| Ethics oversight | Tissue samples for the reference individual, and the male testis, were collected opportunistically when individuals were euthanised for medical purposes. Ear biopsies are collected as part of the metapopulation routine monitoring programs, or during targeted trapping and capture events, that they conducted in accordance with the standard operating procedures for each organization. These management samples were shared with us as part of a study plan approved by representatives from the participating ZAA facilities, the AWC, the Australian Museum, the University of Sydney, and the Greater Bilby National Recovery Team Metapopulation Committee. |

Note that full information on the approval of the study protocol must also be provided in the manuscript.

# Plants

| | |
|---|---|
| Seed stocks | *Report on the source of all seed stocks or other plant material used. If applicable, state the seed stock centre and catalogue number. If plant specimens were collected from the field, describe the collection location, date and sampling procedures.* |
| Novel plant genotypes | *Describe the methods by which all novel plant genotypes were produced. This includes those generated by transgenic approaches, gene editing, chemical/radiation-based mutagenesis and hybridization. For transgenic lines, describe the transformation method, the number of independent lines analyzed and the generation upon which experiments were performed. For gene-edited lines, describe the editor used, the endogenous sequence targeted for editing, the targeting guide RNA sequence (if applicable) and how the editor was applied.* |
| Authentication | *Describe any authentication procedures for each seed stock used or novel genotype generated. Describe any experiments used to assess the effect of a mutation and, where applicable, how potential secondary effects (e.g. second site T-DNA insertions, mosiacism, off-target gene editing) were examined.* |

