## [Peer Review File · Nature Ecology & Evolution]

Peer Review Information

Journal: Nature Ecology & Evolution

Manuscript Title: Extant and extinct bilby genomes combined with Indigenous knowledge improve conservation of a unique Australian marsupial

Corresponding author name(s): : Carolyn Hogg

Editorial Notes:

Reviewer Comments & Decisions:

Decision Letter, initial version:

19th February 2024

Dear Carolyn,

Your manuscript entitled "Extant and extinct bilby genomes combined with Indigenous knowledge improve conservation of a unique Australian marsupial" has now been seen by three reviewers, whose comments are attached. The reviewers have raised a number of concerns which will need to be addressed before we can offer publication in Nature Ecology & Evolution. We will therefore need to see your responses to the criticisms raised and to some editorial concerns, along with a revised manuscript, before we can reach a final decision regarding publication.

Regarding comments from Reviewer# 2, we don't feel strongly about removing some topics but we do agree that they need to be better integrated as the reviewer suggests. We also felt that the Abstract is very jumpy between content at the moment and it needs to provide a better explanation for the "improve conservation" words in the title. We think it is important that the abstract makes clear that you have evaluated existing conservation programs, not just that you are using the genomic and Indigenous knowledge to propose management.

As for format, we estimate that your manuscript is currently about 4000, which is fine but it has too many display items. We have a limit of 6 but you can easily convert some to Extended Data figures.

We therefore invite you to revise your manuscript taking into account all reviewer and editor comments. Please highlight all changes in the manuscript text file in Microsoft Word format.

2* If you have not done so already please begin to revise your manuscript so that it conforms to our Article format instructions at <http://www.nature.com/natecolevol/info/final-submission>. Refer also to any guidelines provided in this letter.

[REDACTED]

Nature Ecology & Evolution is committed to improving transparency in authorship. As part of our efforts in this direction, we are now requesting that all authors identified as 'corresponding author' on published papers create and link their Open Researcher and Contributor Identifier (ORCID) with their account on the Manuscript Tracking System (MTS), prior to acceptance. ORCID helps the scientific community achieve unambiguous attribution of all scholarly contributions. You can create and link your ORCID from the home page of the MTS by clicking on 'Modify my Springer Nature account'. For more information please visit www.springernature.com/orcid.

[REDACTED]

Reviewer expertise:

Reviewer #1: genome sequencing, bioinformatics for conservation genomics

Reviewer #2: evolutionary genomics, conservation

Reviewer #3: land management and conservation in Australia, including Indigenous knowledge

2Reviewers' comments:

Reviewer #1 (Remarks to the Author):

This article is a wonderful observation of Ninu and Yallara, connecting genomics to real-world conservation impacts and comparing it with an extinct but closely related species. The history of the management (zoos, fenced sanctuaries, and islands) are important to understanding the impacts and dynamics of the population. Connecting Ninu and Yallara, genomics, and expounding on the cultural significance and connections is wonderful as well. I found the paper a joy to read and quite informative.

The paper does touch on multiple topics, which can be a little difficult to follow, which should not impact the paper itself.

Major Notes

Genome-informed conservation, Pg 7, Line 204: While the small sample size is provided upfront, some discussion of expected changes if more individuals were sequenced is called for, perhaps with a nod to the relatedness between these individuals.

pg 10, line 267: I'd be curious how these 35 and 4 markers were chosen, and what MassARRAY is, briefly, should be explained. Some description of how the cost of this sort of sequencing makes it more accessible to conservation practitioners should be included, here or in the Conclusion. Not exact costs, but why it should be performed.

Are there any plans for this work to continue, or to change? It would be good to see brief discussion of that in the conclusion.

pg 15 Line 333: Is this where the GWAS mentioned in the methods is from?

Extended supplemental Management Recommendations is good, and perhaps should be highlighted from within the article itself more?

GitHub link is a repository that has had no updates in 4 years. Is this still correct? Perhaps best to turn it into a DOI to give a specific link, as the repo looks generic for the lab. Keeping as much code reproducible as possible is important for future conservation work, as it provides a guide for other researchers coming into this space.

Minor Notes

Table 1 is missing BUSCO and % Gaps for Short-tailed opossum.
BUSCO version should be consistent, thus should be re-run for Tasmanian devil using v5.3.2.
Significant digits for GC content should match for Tasmanian Devil (36.00, if it truly is exact).

3pg 10, starting at line 245: This paragraph is very dense, could it be split? This is an incredibly minor note, so if not addressed I don't think it should be a stopper.

Figure 3: The small text is difficult to read. Perhaps this could be more focused, and some plots moved to supplemental?

pg 15, line 330: The jumping between the amount of SNPs and data is abrupt. I wonder if names for each dataset (Field Monitoring (39 SNPs)), and likewise for 3,858 SNPs.

Fig 4 a: Text is very difficult to read. Some points do not have text associated, but I don't see where the cutoff is.

Fig 4 c. Should be a table, probably supplemental.

Fig 5 b. Absolutely lovely.

Extended Data Table 2. Thank you for making it clear when BUSCO is used for proteins or transcriptomes.

Online Methods line 44: Why are there two versions of BUSCO being used?

Reviewer #2 (Remarks to the Author):

This is a very comprehensive paper, addressing several relevant topics related to a biologically interesting and threatened species. Some of these topics are also addressed in its extinct sister-species, whose genome sequence is also presented.

On the one hand, the paper is a 'tour de force', encompassing a wide variety of approaches to characterize the target species' genome and to address multiple questions. On the other hand, this diversity is also challenging in terms of showing cohesiveness among the various components of the paper. Some of them are more clearly presented than others, and the shift from one topic to the next is not always clear.

Although one can see that there is a common underlying theme (the bilby's genome), the paper seems to comprise multiple studies, addressing in some cases rather disparate topics, and with sometimes tenuous connections among them.

So, my overall assessment is that the paper does present sufficient robustness for potential publication in NEE, as it contributes a very substantial amount of data on a relevant topic, and presents a variety of analyses that are mostly well conducted and explained (especially if one accesses the detailed explanations provided in the supplementary methods).

At the same time, my main concern is that the cohesiveness among the topics should be improved.

4One possibility is to remove some of the topics (i.e., publishing them as separate papers), so as to produce a more focused 'story', and to have more space to go into more depth on the focused topics. In some topics addressed in the current manuscript, there is a lack of depth, with results and interpretations being presented rather cursorily, which I assume is a consequence of trying to include too many (and too varied) topics in a single paper, without choosing some of them to comprise the main story. I defer to the editor to indicate whether this (excluding some topics) is an interesting option for this paper.

If the editor and authors deem that no topic should be excluded, which is also defensible, I would still recommend that some of them be chosen for presentation in more depth in the main text, and thus comprise the main 'story' of the paper, while others are briefly mentioned in the main text (as additional findings) and presented only in the supplements, which are already extensive, but do present some of the methods and results more clearly than the main text.

Regardless of which topics are kept in the main text, I recommend that their order be reconsidered, so as to provide a clearer flow to the reader. For example, if all the current topics were kept, I would suggest (i) beginning with an abridged version of the sections describing the genome features and broader evolutionary comparisons with other marsupials and eutherians; then (ii) moving to analyses addressing the evolutionary history of Ninu and Yallara; then (iii) describing the analyses inferring adaptive differentiation between the temperate and semi-arid Ninu populations (which require additional detail in the main text); and then (iv) devoting most attention to what seems to be the main highlight of the study, which are the population-level analyses and genome-informed conservation. I think this order would improve clarity and readability of the paper, so that the readers can better appreciate the amount of work put into it and the interesting and relevant findings that it reports.

I list below some specific concerns:

Lines 60-62: 'Together, we demonstrate the holistic value of genomics in addressing key Indigenous and conservation management questions for ongoing conservation efforts.'

- I think the 'Indigenous' element is a bit oversold here and elsewhere in the paper, considering its current structure. While I can see that one of the addressed topics is related to Indigenous management questions, which is very good, several others are not. This is connected to the criticism I mentioned above, of addressing too many topics. One possibility is to tone down the 'Indigenous' component of the paper in this sentence and others. The other is to remove or reduce the emphasis on some of the other topics (e.g., broader evolutionary comparisons), and to expand the focus on the conservation and management components, which would then match this statement more clearly.

Line 136: '...which consists of insects, insect larvae, termites, seeds, bulbs'

- Termites are insects too.

Line 140: 'bandicoots, belong to the order Peramelemorphia, and unlike other marsupials have a short-lived chorioallantoic placentas as well as a choriovitelline placenta.'

5- It should read 'a short-lived chorioallantoic placenta'.

Lines 172-174: 'The Yallara genome assembly is 3.50 Gb in size (6,329,012 contigs; 19.74% gaps; 85.5% complete mammalian BUSCOs; Table 1)'

- How was this Yallara assembly produced? It is mentioned here for the first time in the main text. I recommend that it be described prior to this point, as was done for the Ninu.

Lines 184-186: 'The phylogenetic relationship among the Ninu (N = 6) and Yallara (N = 2) individuals were inferred using the mitochondrial genomes extracted from the final BAM files (Figure 1C)'

- Why only mtDNA? Why not use whole-genome data to investigate their divergence? Later in the paper I could see the SMC-based analyses of their divergence, which are informative, but I think the paper should make more use of (or at least mention more clearly) the nuclear data for this purpose.

- Why only two Yallara individuals, since the previous sentence indicates that four were sequenced?

Lines 186-188: 'For the high coverage dataset (total 2,146 variants), PC1 splits the Yallara and Ninu explaining 61.6% of the variance (Figure 1D). PC2 splits the Ninu samples into semi-arid and temperate samples, explaining a further 10.9% of the variance observed (Figure 1D).'

- It would be good to define what the high-coverage dataset is before its results are presented. Also, the percentages of the variance explained by these PCs are different in the text and the figure, so this should be checked.

Lines 212-213: 'The semi-arid Ninu population was originally sourced from the last remaining Ninu population at Astrebla Downs, Queensland (24.20°S, 140.55°E)'

- Is this shown on any map? I have not found it. It would be useful to see this in Fig 1A, for example.

Lines 220-221: 'small founder sizes and captive breeding, could be partially attributed to their shorter generation time (hence higher recombination).'

- This is a bit unclear and perhaps can be explained in more detail. Shorter generation times would not affect recombination rates per se, but perhaps the authors mean the total amount of recombination per absolute time (e.g. per year). If so, this can be made explicit. At the same time, a shorter generation time also implies more inbreeding and more drift per absolute time, which would be expected to have the opposite effect. So, this is something that can be considered and discussed in a bit more detail. Perhaps this is actually indicating an interesting property (i.e. recombination-related) of this genome, which is something that may be worth investigating in the future.

- Another, more technical aspect, is to ascertain that the method employed to call ROHs here is fully comparable to those applied in previous papers. I think the authors can check it carefully and mention this explicitly, if it is the case.

Lines 330-335: 'Other biological insights from the genome data.

A total of 3,858 SNPs that met our criteria were common across all three association analyses (Chi-square association, Fisher's test, FST outlier test). We identified 339 unique genes between semi-arid and temperate individuals (Figure 4A; a full list of GO terms and associated genes is given in Table S4). As only two individuals had high sequencing coverage for the Yallara (Table S1), association analyses could not be undertaken.'

- This section is not sufficiently connected to the previous sections, refers to analyses that have not been clearly presented up to this point (in the main text), and that are not clear to the reader. What are the common criteria? Where do these SNPs come from? Are they from the RRS set presented in an earlier section? If so, it should be clearly stated. What are the association analyses? Association with what? I assume it is with the two types of environments, but it should be presented explicitly. This set of analyses needs to be explained more clearly, so that the reader can understand their results.

- Moreover, the topic should be introduced clearly, with an informative title, instead of 'Other biological insights from the genome data'. It is unclear whether this title refers to this specific section (which appears to focus on the potential adaptive differentiation between semi-arid and temperate populations), or to the whole set of sections that follows from this point onwards. So, this needs to be revised and clarified.

Figure 1:

- Panels A and B) I recommend showing (at least approximately) the sample localities for the individuals that were analysed. This is relevant in terms of comparing the temperate vs. semi-arid Ninu populations, for example.

- Panel C) I recommend adding estimates of divergence dates to this phylogenetic tree.

- Panel D) Check percentages of the variance explained, as they don't match the text. Also, the acronym WGR must be defined in the legend itself, so the reader does not need to look for it in the text.

Figure 2. 'The coloured bars represent the proportion of the genome in ROH>100kb (FROH>100 k; red), ROH>500kb (FROH>500 kb; green), and ROH>1 Mb (FROH>1 Mb; blue), while the crosses represent observed heterozygosity'

- Are these ROH sizes correct? If so, it implies that the bars are not cumulative, since the FROH>1 Mb is contained within the FROH > 500 kb, which in turn is contained within the FROH> 100kb. This is not the usual way to depict this, since it is informative to assess both the proportion of the genome within a given size range and the total FROH, which would be the sum of the different categories. So, I recommend revising this figure to show non-overlapping ROH size ranges.

Figure 3: 'A) Map showing all sampling locations and the Kiwirrkurra Community (black box)'

7-Is the Kiwirrkurra Community the black box? Or is it the star within the black box? The inset map (panel B) indicates the latter.

Reviewer #3 (Remarks to the Author):

This manuscript offers a good perspective on right-way science by supporting Indigenous-led research questions into the overall objectives of the study. Highlighting the importance and benefits of empowering Indigenous people in species management, while promoting the value of Indigenous Knowledge. I applaud the effort of the authors to include Indigenous perspectives in this otherwise highly technical manuscript, it will be an exemplar of inclusion of right-way science once published.

Please note, I am only providing comment on the Indigenous Knowledge inclusion in this manuscript. I make these comments understanding the constraints of word limits and NEE style – however, if possible, I think the authors could further strengthen the inclusion of Indigenous Knowledge by:

- acknowledging cultural landscape for Kiwirrkurra in the online methods
- including a clear statement about other Traditional Owner groups values, Indigenous Knowledge, language and loss of knowledge due to species decline. As has been demonstrated in DCCEEW 2023, Recovery Plan for the Greater Bilby (*Macrotis lagotis*), Department of Climate Change, Energy, the Environment and Water, Canberra. CC BY 4.0
- Line 180 and 213 (and others) authors need to carefully consider the use of 'Ninu' when referring to populations of Bilby outside of Kiwirrkurra as this could offend other Traditional Owner groups where this samples were collected.
- Line 95. The concept of translocation for many Aboriginal groups can be culturally sensitive – a statement to this effect could be added to this manuscript.
- Lines 147-153, it is likely that Indigenous Knowledge would support this concept of 'boom and bust'. If possible and ICIP is supported an appropriate 'Ninu' story could be shared. This would demonstrate the two Knowledge systems side by side.
- Line 76. Consider replacing the term 'ornaments' with 'cultural artefact' or 'used in cultural practice'.
- Line 270 and Line 273 both community rangers and Indigenous rangers are used throughout; I suggest consistency using Indigenous rangers.
- Lines 498-500, depending on how the above feedback is incorporated, this sentence will need revision.

*****END*****

Author Rebuttal to Initial comments

Reviewer #1 (Expertise: genome sequencing, bioinformatics for conservation genomics):

This article is a wonderful observation of Ninu and Yallara, connecting genomics to real-world

8conservation impacts and comparing it with an extinct but closely related species. The history of the management (zoos, fenced sanctuaries, and islands) are important to understanding the impacts and dynamics of the population. Connecting Ninu and Yallara, genomics, and expounding on the cultural significance and connections is wonderful as well. I found the paper a joy to read and quite informative.

The paper does touch on multiple topics, which can be a little difficult to follow, which should not impact the paper itself.

Response: We have adjusted the text throughout as suggested by Reviewer 2 to improve flow in the manuscript and easier to follow.

Major Notes

Genome-informed conservation, Pg 7, Line 204: While the small sample size is provided upfront, some discussion of expected changes if more individuals were sequenced is called for, perhaps with a nod to the relatedness between these individuals.

Response: The bold text has been added at lines 199-203 for clarification: *“As expected from the significant inbreeding observed in the reduced representation sequencing data (Table 1), the six Ninu from the temperate island population (Thistle Island) had lower heterozygosity and generally higher ROH-based inbreeding coefficients (FROH) than those from the semi-arid population (Fig. 2C; Table S2), even though unrelated individuals were selected for resequencing.”*

pg 10, line 267: I'd be curious how these 35 and 4 markers were chosen, and what MassARRAY is, briefly, should be explained. Some description of how the cost of this sort of sequencing makes it more accessible to conservation practitioners should be included, here or in the Conclusion. Not exact costs, but why it should be performed.

Response: To clarify the questions raised by the reviewer, we have added the following text at lines 245-248: *“Using RRS SNPs aligned to the reference genome, a MassARRAY (mass spectrometry with end-point PCR) panel of 35 autosomal and four sex-linked markers was developed (Supplementary Note 2.5). SNP loci were selected based on high minor allele frequency (>0.30) in populations across the species distribution and with high reproducibility.”*

The following text has been added to the conclusion at line 416-417: *“.....wild populations in the future due to the relative ease of scat collection and low cost compared to trapping and tissue sampling”.*

Are there any plans for this work to continue, or to change? It would be good to see brief discussion of that in the conclusion.

Response: The following text has been added to the conclusion at lines 419-422: *“Next steps for the scat protocol are to continue to use it for the bilby monitoring program in the Pilbara, and undertake a nationwide scat survey in collaboration with Indigenous communities, not-for-profit organisations, and Indigenous school groups to provide more comprehensive comparison between the metapopulation and wild remnant populations.”*

pg 15 Line 333: Is this where the GWAS mentioned in the methods is from?

Response: We have added the following text at Lines 286-288 to provide clarity: *“To assess potential allele frequency differences between semi-arid and temperate individuals, we performed a genome-wide association analysis (GWAS) using the 12 resequenced Ninu genomes.”*

Extended supplemental Management Recommendations is good, and perhaps should be highlighted from within the article itself more?

Response: Thank you, we have included the following text at line 270-272: *“These recommendations included maximising genetic diversity and value of the metapopulation, sourcing wild bilbies, biobanking genetic samples and using the scat method to undertake a nationwide survey.”*

GitHub link is a repository that has had no updates in 4 years. Is this still correct? Perhaps best to turn it into a DOI to give a specific link, as the repo looks generic for the lab. Keeping as much code reproducible as possible is important for future conservation work, as it provides a guide for other researchers coming into this space.

Response: All Github links in the manuscript are provided in addition to citations for software used in the analysis (e.g. PacBio Improved Phase Assembler, PreText Suite etc.) and are operated by other research groups who developed the software. It is not within our capacity to create DOIs for these Github links, however the full citation is provided where the software has been published in a journal.

We have updated the Code availability statement in the paper to read: *“The code used to select SNPs to design the custom MassARRAY scat genotyping assay is provided as Supplementary Text File. All other analyses used standard software and scripts as described in the Methods and Supplementary.”*

Minor Notes

Table 1 is missing BUSCO and % Gaps for Short-tailed opossum. BUSCO version should be consistent, thus should be re-run for Tasmanian devil using v5.3.2. Significant digits for GC content should match for Tasmanian Devil (36.00, if it truly is exact).

Response: The BUSCO and % gaps for the opossum have now been included in Extended Table 2 (previously Table 1). We have re-run the Tasmanian devil BUSCO as per the reviewer's request with Version 5.3.2 and updated the significant figures across the table for consistency.

pg 10, starting at line 245: This paragraph is very dense, could it be split? This is an incredibly minor note, so if not addressed I don't think it should be a stopper.

Response: We have split this paragraph at Line 233 – separating out the section on heterozygosity and the changes observed in the translocated populations, from the rest of the population genetic data i.e. inbreeding, allelic richness and effective population sizes.

Figure 3: The small text is difficult to read. Perhaps this could be more focused, and some plots moved to supplemental?

Response: Figure 3 has been re-created to keep it more focused and easier to read. The PCoAs have been moved to the supplementary section 2.6 (Fig. S10) and the maps made bigger to improve the quality of the text.

pg 15, line 330: The jumping between the amount of SNPs and data is abrupt. I wonder if names for each dataset (Field Monitoring (39 SNPs)), and likewise for 3,858 SNPs.

Response: In order to clarify what data we are discussing in this section, we have added the following text at Lines 286-288: *“To assess potential adaptive allele frequency differences between semi-arid and temperate individuals, we performed a genome-wide association analysis (GWAS) using the 12 resequenced Ninu genomes.”*

Fig 4 a: Text is very difficult to read. Some points do not have text associated, but I don't see where the cutoff is.

Response: Figure 4A has been moved to the extended data section of the paper and is now Extended Data Fig. 2. The figure visualises the results of Revigo semantic reduction of the GO terms. Revigo calculates dispensability, where a higher dispensability value represents GO term semantic redundancy

11with respect to the more relevant cluster label. For ease of visualisation, only some GO term labels are displayed. These were filtered by a dispensability value less than the median dispensability in our dataset (0.074). We have added the following text to the caption at Lines 1141-1144: *“Terms with a dispensability (measure of semantic redundancy) value lower than the median (0.074) are labelled. A full list of enriched GO terms is provided in Table S4. Uniqueness is calculated as 1 minus the average semantic similarity of a term to all other terms.”*

Fig 4 c. Should be a table, probably supplemental.

Response: The table in the Figure has been moved to the extended data section and is now Extended Data Table 5.

Fig 5 b. Absolutely lovely.

Response: Thank you, we like it too. This is now Fig 4.

Extended Data Table 2. Thank you for making it clear when BUSCO is used for proteins or transcriptomes.

Response: We thank the reviewer for their comment.

Online Methods line 44: Why are there two versions of BUSCO being used?

Response: This is because we undertook assessment at different times of the genome completeness. To make this clearer we have added the following text in the online methods at Lines 475-479: *“Early assessments of assembly completeness were conducted with Benchmarking Universal Single-Copy Orthologs (BUSCO) v3.0.2b (RRID:SCR_015008)¹⁸ (mammalia_odb9; 4,104 genes). Subsequently, BUSCOMP v1.1.2 (RRID:SCR_021233)¹⁹ and BBTools v38.73 (RRID:SCR_016968)²⁰ were used to generate general assembly statistics and assembly completeness was assessed using BUSCO v5.4.4¹⁸ (mammalia_odb10; 9,226 genes).”*

Reviewer #2 (evolutionary genomics, conservation):

This is a very comprehensive paper, addressing several relevant topics related to a biologically interesting and threatened species. Some of these topics are also addressed in its extinct sister-species, whose genome sequence is also presented.

On the one hand, the paper is a 'tour de force', encompassing a wide variety of approaches to characterize the target species' genome and to address multiple questions. On the other hand, this diversity is also challenging in terms of showing cohesiveness among the various components of the paper. Some of them are more clearly presented than others, and the shift from one topic to the next is not always clear.

Although one can see that there is a common underlying theme (the bilby's genome), the paper seems to comprise multiple studies, addressing in some cases rather disparate topics, and with sometimes tenuous connections among them.

So, my overall assessment is that the paper does present sufficient robustness for potential publication in NEE, as it contributes a very substantial amount of data on a relevant topic, and presents a variety of analyses that are mostly well conducted and explained (especially if one accesses the detailed explanations provided in the supplementary methods).

At the same time, my main concern is that the cohesiveness among the topics should be improved. One possibility is to remove some of the topics (i.e., publishing them as separate papers), so as to produce a more focused 'story', and to have more space to go into more depth on the focused topics. In some topics addressed in the current manuscript, there is a lack of depth, with results and interpretations being presented rather cursorily, which I assume is a consequence of trying to include too many (and too varied) topics in a single paper, without choosing some of them to comprise the main story. I defer to the editor to indicate whether this (excluding some topics) is an interesting option for this paper.

If the editor and authors deem that no topic should be excluded, which is also defensible, I would still recommend that some of them be chosen for presentation in more depth in the main text, and thus comprise the main 'story' of the paper, while others are briefly mentioned in the main text (as additional findings) and presented only in the supplements, which are already extensive, but do present some of the methods and results more clearly than the main text.

Regardless of which topics are kept in the main text, I recommend that their order be reconsidered, so as to provide a clearer flow to the reader. For example, if all the current topics were kept, I would suggest (i) beginning with an abridged version of the sections describing the genome features and broader evolutionary comparisons with other marsupials and eutherians; then (ii) moving to analyses addressing the evolutionary history of Ninu and Yallara; then (iii) describing the analyses inferring adaptive differentiation between the temperate and semi-arid Ninu populations (which require additional detail in the main text); and then (iv) devoting most attention to what seems to be the main highlight of the study, which are the population-level analyses and genome-informed conservation. I

think this order would improve clarity and readability of the paper, so that the readers can better appreciate the amount of work put into it and the interesting and relevant findings that it reports.

Response: Although we understand this paper contains a vast array of information, we wanted to include it all to show the depth and scope of the possibilities of developing and using high-quality genomic resources for conserving a threatened species and improving knowledge of their biology. This comprehensive study arose from a need to develop genomic tools that could be used to inform the conservation management of this threatened species, both with the Indigenous rangers who manage the species in the wild but also the managed metapopulation. This is why the “Genome-informed conservation” section comes after the abridged version of the “Genome landscape” section. These tools are not only for understanding population genetics, but also some of the unique adaptations and biology of the species, that may be impacted by mixing bloodlines/localities at the continental scale. As bilbies have such unique biology, we used this opportunity to investigate the genomic basis of their unusual chromosomal system, chorioallantoic placenta and their immune system.

To address the reviewers concerns, we have added a series of linking sentences throughout the manuscript to improve the flow; made changes to the abstract as recommended by the Editorial team to clearly show that we have evaluated existing conservation programs, and are not just using genomic and Indigenous knowledge to propose management actions; and as per Reviewer 1, Reviewer 2 and the Editor’s suggestions reduced the number of display items and moved others to the extended data. By doing so, we feel we have better highlighted the conservation story, so it is now clearer that this was where this comprehensive study started.

I list below some specific concerns:

Lines 60-62: ‘Together, we demonstrate the holistic value of genomics in addressing key Indigenous and conservation management questions for ongoing conservation efforts.’

- I think the ‘Indigenous’ element is a bit oversold here and elsewhere in the paper, considering its current structure. While I can see that one of the addressed topics is related to Indigenous management questions, which is very good, several others are not. This is connected to the criticism I mentioned above, of addressing too many topics. One possibility is to tone down the ‘Indigenous’ component of the paper in this sentence and others. The other is to remove or reduce the emphasis on some of the other topics (e.g., broader evolutionary comparisons), and to expand the focus on the conservation and management components, which would then match this statement more clearly.

Response: Both the Editor and Reviewer 3 have asked us to further highlight the Indigenous knowledge and cultural landscape for this species. As a result of all Reviewer and Editor comments, we have tried to reduce the emphasis on the other unique biological aspects of the species and make the conservation story clearer. This has been achieved through changes to the Abstract (lines 49-53 and 62-64), inclusion of cultural landscape in the Methods (lines 559-567), acknowledgment of species' Indigenous names (Lines 76-77) and cultural importance (lines 79-83), the cultural conflict with translocations (Lines 103-104), and clarification on the purpose of the study (Lines 140-147).

Line 136: '...which consists of insects, insect larvae, termites, seeds, bulbs'

- Termites are insects too.

Response: Deleted the word "termites"

Line 140: 'bandicoots, belong to the order Peramelemorphia, and unlike other marsupials have a short-lived chorioallantoic placentas as well as a choriovitelline placenta.'

- It should read 'a short-lived chorioallantoic placenta'.

Response: Deleted the "s" off placenta to read as the reviewer recommended.

Lines 172-174: 'The Yallara genome assembly is 3.50 Gb in size (6,329,012 contigs; 19.74% gaps; 85.5% complete mammalian BUSCOs; Table 1)'

- How was this Yallara assembly produced? It is mentioned here for the first time in the main text. I recommend that it be described prior to this point, as was done for the Ninu.

Response: The following text has been added at Lines 161-164: "*A Yallara genome assembly was generated from a skull sample collected in 1898 and sequenced with short-read (Illumina) sequencing (male NMVC7087; Table S1). We used the bwa aln algorithm to align reads to the Ninu genome (version 1.9; Supplementary Section 1.7).*"

Lines 184-186: 'The phylogenetic relationship among the Ninu (N = 6) and Yallara (N = 2) individuals were inferred using the mitochondrial genomes extracted from the final BAM files (Figure 1C)'

- Why only mtDNA? Why not use whole-genome data to investigate their divergence? Later in the paper I could see the SMC-based analyses of their divergence, which are informative, but I think the paper should make more use of (or at least mention more clearly) the nuclear data for this purpose.

15Response: We thank the reviewer for identifying this miscommunication. In addition to the mitogenome analysis, we also performed two separate PCAs on whole genome datasets to investigate the divergence between the Ninu and Yallara, and within the semi-arid and temperate Ninu. Both the mitogenome and the whole genome datasets clearly confirmed these divergences, but the way the section was written did not emphasise this result.

We have now edited the section to communicate the consistent results from the mitochondrial and whole genome datasets more clearly (Lines 172-182): *“The phylogenetic relationship among the Ninu and Yallara individuals was confirmed using both full mitogenomes and whole genome nuclear data. Mitogenomes were extracted from the final BAM files generated for the high coverage Ninu (N = 6) and Yallara (N = 2) individuals (Table S1), and a phylogenetic tree was constructed using both Maximum Likelihood and Bayesian methods (Fig. 1C). PCAs were also generated for the whole genome datasets, which both confirmed the mitogenome divergence results (Figure 1D, Supplemental Note 1.7, Figure S7). For the dataset including only high coverage individuals (total 2,787 variants), PC1 splits the Yallara and Ninu explaining 61.6% of the variance (Fig. 1D). PC2 splits the Ninu samples into semi-arid and temperate samples, explaining a further 11% of the variance observed (Fig. 1D) consistent with the mitogenome results (Fig. 1C).”*

- Why only two Yallara individuals, since the previous sentence indicates that four were sequenced?

Response: Although there were 4 Yallara individuals sequenced, only 2 of those individuals yielded high enough coverage genomes to be used in some analyses (see Table S1, now identified with asterisk). However, we confirmed the low coverage Yallara were genetically similar to the high coverage Yallara by performing a PCA where the 2 low coverage individuals were projected on to the high coverage individuals (see Figure S7). We have now edited the sections of the manuscript (Lines 172-182; see above) and supplementary material where we discuss our PCA results so this is more clear (Lines 380-385 Supplementary).

Lines 186-188: ‘For the high coverage dataset (total 2,146 variants), PC1 splits the Yallara and Ninu explaining 61.6% of the variance (Figure 1D). PC2 splits the Ninu samples into semi-arid and temperate samples, explaining a further 10.9% of the variance observed (Figure 1D).’

- It would be good to define what the high-coverage dataset is before its results are presented. Also, the percentages of the variance explained by these PCs are different in the text and the figure, so this should be checked.

Response: We thank the reviewer for pointing out this issue. We made a mistake and included the projected PCA rather than the high coverage PCA in the main text. We have now edited this section and included the correct PCA images in the main text and supplementary sections.

Lines 212-213: 'The semi-arid Ninu population was originally sourced from the last remaining Ninu population at Astrebla Downs, Queensland (24.20°S, 140.55°E)'

- Is this shown on any map? I have not found it. It would be useful to see this in Fig 1A, for example.

Response: The location of Astrebla Downs has been added to Figure 1A, and added to the text at Line 209: ".....Astrebla Downs, Queensland (24.20°S, 140.55°E; Fig. 1A).

Lines 220-221: 'small founder sizes and captive breeding, could be partially attributed to their shorter generation time (hence higher recombination)'

- This is a bit unclear and perhaps can be explained in more detail. Shorter generation times would not affect recombination rates per se, but perhaps the authors mean the total amount of recombination per absolute time (e.g. per year). If so, this can be made explicit. At the same time, a shorter generation time also implies more inbreeding and more drift per absolute time, which would be expected to have the opposite effect. So, this is something that can be considered and discussed in a bit more detail. Perhaps this is actually indicating an interesting property (i.e. recombination-related) of this genome, which is something that may be worth investigating in the future.

Response: We agree with the reviewer that without additional evidence it is difficult to tease apart these various potential factors; and is certainly something interesting to explore further, but outside the scope of our study. We have clarified the text at Lines 215-220, by adding in the bold text below: "*The fewer long ROHs in Ninu, despite their history of small founder sizes and captive breeding, could be partially attributed to **their boom- bust demographic history**, and/or their shorter generation time (hence higher **per-year recombination rate**) and potentially higher substitution rate than those of larger mammals²³. **Further work could tease apart the influence of the species' demography and intrinsic biological characteristics on the ROH distribution.***"

- Another, more technical aspect, is to ascertain that the method employed to call ROHs here is fully comparable to those applied in previous papers. I think the authors can check it carefully and mention this explicitly, if it is the case.

Response: The methods and parameters to compute ROH vary across studies due to differences in genome quality, genome size and SNP density (Meyermans et al., 2020; Duntsch et al. 2021). However, the use of PLINK to characterise ROH remains a common approach. We chose PLINK parameters following recommendations in Kardos et al. (2015), Ceballos et al. (2018) and Meyermans et al. (2020) - this has now been explicitly stated in the manuscript at Lines 525-527: *“We chose PLINK settings for the ROH analysis of this high-density SNP dataset following recommendations in Kardos et al.⁸⁹, Ceballos et al.⁹⁰ and Meyermans et al.⁹¹.”*

We note that most PLINK settings become unimportant when marker density is very high (e.g. the minimum density parameter), as is the case for the bilby genomes. We have also added the results of additional analyses performed under various PLINK parameter combinations to ascertain the sensitivity of the results to the choice of parameter, and to enable a wider range of comparisons (see Table S9). The text has been adjusted accordingly at Lines 532-533: *“We reran these ROH analyses applying various parameter combinations to ascertain the sensitivity of the results to the choice of parameter (Table S9).”*

We have explicitly stated the latter point in the methods and have provided relevant references. Note that we present different categories of FROH (i.e. FROH>100kb, FROH>500kb and FROH>1Mb) to enable comparisons to other studies that present varying FROH categories.

- Ceballos, F. C., Hazelhurst, S., & Ramsay, M. (2018). Assessing runs of Homozygosity: a comparison of SNP Array and whole genome sequence low coverage data. *BMC genomics*, 19, 1-12.
- Duntsch, L., Whibley, A., Brekke, P., Ewen, J. G., & Santure, A. W. (2021). Genomic data of different resolutions reveal consistent inbreeding estimates but contrasting homozygosity landscapes for the threatened Aotearoa New Zealand hihi. *Molecular Ecology*, 30(23), 6006-6020.
- Kardos, M., Luikart, G., & Allendorf, F. W. (2015). Measuring individual inbreeding in the age of genomics: marker-based measures are better than pedigrees. *Heredity (Edinb)*, 115(1), 63–72.
- Meyermans, R., Gorssen, W., Buys, N., & Janssens, S. How to study runs of homozygosity using PLINK? A guide for analyzing medium density SNP data in livestock and pet species. *BMC genomics* 21: 1-14 (2020).

Lines 330-335: ‘Other biological insights from the genome data. A total of 3,858 SNPs that met our criteria were common across all three association analyses (Chi-square association, Fisher’s test, FST outlier test). We identified 339 unique genes between semi-arid and temperate individuals (Figure 4A; a full list of GO terms and associated genes is given in Table S4). As only two individuals had high sequencing coverage for the Yallara (Table S1), association analyses could not be undertaken.’

- This section is not sufficiently connected to the previous sections, refers to analyses that have not been clearly presented up to this point (in the main text), and that are not clear to the reader. What are the common criteria? Where do these SNPs come from? Are they from the RRS set presented in an earlier section? If so, it should be clearly stated. What are the association analyses? Association with what? I assume it is with the two types of environments, but it should be presented explicitly. This set of analyses needs to be explained more clearly, so that the reader can understand their results.

Response: The following text has been added to this section at Lines 288-289 to provide clarity. *“To assess potential adaptive allele frequency differences between semi-arid and temperate individuals, we performed a genome-wide association analysis (GWAS) using the 12 resequenced Ninu genomes.”*

“A total of 3,858 SNPs that met our criteria (bi-allelic SNPs, no missing data, minor allele frequency >0.05) were common across all three association analyses (Chi-square association, Fisher’s test, FST outlier test). As a result, we identified 339 enriched genes between semi -arid and temperate individuals (Figure 4A; a full list of GO terms and associated genes is given in Table S4). As only two individuals had high sequencing coverage for the Yallara (Table S1), association analyses could not be undertaken.”

- Moreover, the topic should be introduced clearly, with an informative title, instead of ‘Other biological insights from the genome data’. It is unclear whether this title refers to this specific section (which appears to focus on the potential adaptive differentiation between semi-arid and temperate populations), or to the whole set of sections that follows from this point onwards. So, this needs to be revised and clarified.

Response: For clarification we have changed the title of the section and added a linking sentence to align better with the scope of the project at Lines 281-288:

“Unique biological insights from the Ninu genome data

Further to our primary aim of using genetic data to assess and inform current and future conservation management, we used the resources generated in this study alongside comparative genomics approaches to explore the genomic basis of the Ninu’s unique adaptations.”

An additional heading has been created for the GWAS section titled *“Differences between semi-arid and temperate individuals”* to provide clarity.

Figure 1

- Panels A and B) I recommend showing (at least approximately) the sample localities for the individuals that were analysed. This is relevant in terms of comparing the temperate vs. semi-arid Ninu populations, for example.

Response: Apologies, the temperate and semi-arid information is listed in the legend, but the arrows had been deleted from the map in Panel A. Due to the age of the Yallara specimens, their localities are known only from the states in which they were found, these have been named on the maps now.

- Panel C) I recommend adding estimates of divergence dates to this phylogenetic tree.

Response: The divergence times have now been added to the phylogenetic tree.

- Panel D) Check percentages of the variance explained, as they don't match the text. Also, the acronym WGR must be defined in the legend itself, so the reader does not need to look for it in the text.

Response: We thank the reviewer for pointing out this issue. We made a mistake and included the projected PCA rather than the high coverage PCA in the main text. We have now edited this section and included the correct PCA images in the main text and supplementary sections. The acronym has been defined in the legend.

Figure 2. 'The coloured bars represent the proportion of the genome in ROH>100kb (FROH>100 k; red), ROH>500kb (FROH>500 kb; green), and ROH>1 Mb (FROH>1 Mb; blue), while the crosses represent observed heterozygosity'

- Are these ROH sizes correct? If so, it implies that the bars are not cumulative, since the FROH>1 Mb is contained within the FROH > 500 kb, which in turn is contained within the FROH> 100kb. This is not the usual way to depict this, since it is informative to assess both the proportion of the genome within a given size range and the total FROH, which would be the sum of the different categories. So, I recommend revising this figure to show non-overlapping ROH size ranges.

Response: We have now separated the different FROH categories into different columns to display non-overlapping ROH size ranges as suggested by the reviewer.

Figure 3: 'A) Map showing all sampling locations and the Kiwirrkurra Community (black box)'

-Is the Kiwirrkurra Community the black box? Or is it the star within the black box? The inset map (panel B) indicates the latter.

Response: Figure 3 has been re-created as per the recommendations from Reviewer 1 that it was too busy. The PCoAs have been removed to the Supplementary text and the maps made bigger so the text is clearer. Dotted lines have now been added to indicate that the inset in Figure 3B is the zoomed in area of the black box. The Kiwirrkurra Community is the red star, this has been corrected in the figure legend.

Reviewer #3 (land management and conservation in Australia, including Indigenous knowledge):

This manuscript offers a good perspective on right-way science by supporting Indigenous-led research questions into the overall objectives of the study. Highlighting the importance and benefits of empowering Indigenous people in species management, while promoting the value of Indigenous Knowledge. I applaud the effort of the authors to include Indigenous perspectives in this otherwise highly technical manuscript, it will be an exemplar of inclusion of right-way science once published.

Response: Thank you for the kind acknowledgement of our efforts to be inclusive.

Please note, I am only providing comment on the Indigenous Knowledge inclusion in this manuscript. I make these comments understanding the constraints of word limits and NEE style – however, if possible, I think the authors could further strengthen the inclusion of Indigenous Knowledge by:

- acknowledging cultural landscape for Kiwirrkurra in the online methods

Response: The following text has been added to the online methods at Lines 122-130 in acknowledgement of the Kiwirrkurra cultural landscape. *“Scat samples and metadata were collected by the Kiwirrkurra Indigenous rangers. Kiwirrkurra is located in the “tali” (sandhill) country of the Gibson Desert (Fig. 3A) and has been described as the most remote community in Australia⁴⁶. All residents are Pintupi speaking people with close family ties to other Ngaanaytjarra communities. The Kiwirrkurra Indigenous Protected Area is 45,867km² and is Kiwirrkurra Native Title Determination, managed by the Pintupi traditional owners with assistance from Central Desert Native Title Services (CDNTS)⁴⁶. Under their guidance, the Indigenous rangers undertake cultural burning, feral animal and weed control, threatened species monitoring and passing knowledge from elders to young people. Many Kiwirrkurra community members still engage in traditional land-use practices⁴⁶.”*

- including a clear statement about other Traditional Owner groups values, Indigenous Knowledge,

language and loss of knowledge due to species decline. As has been demonstrated in DCCEEW 2023, Recovery Plan for the Greater Bilby (*Macrotis lagotis*), Department of Climate Change, Energy, the Environment and Water, Canberra. CC BY 4.0

Response: Added the following text at lines 79-83: *“Indigenous knowledge, bilby songlines, ceremonies and stories exist across Australia, linking sites and people. Their strong connection to the species continues even in areas where bilbies are now locally extinct. Loss of this Indigenous knowledge and land management practices due to the species’ decline is a recognised threat to the persistence on bilbies in the landscape².”*

- Line 180 and 213 (and others) authors need to carefully consider the use of ‘Ninu’ when referring to populations of Bilby outside of Kiwirrkurra as this could offend other Traditional Owner groups where this samples were collected.

Response: We acknowledge the reviewers point here and as authors debated the use of ‘Ninu’ when we initially wrote the manuscript. Originally, we used a combination of bilby and Ninu to mean the Greater bilby, but this ended up making the manuscript difficult to follow, and we wanted to use an Indigenous name for the species to highlight its cultural value and importance and ensure that Indigenous knowledge of the species is promoted through the scientific literature. This is why we have text in the opening paragraph (lines 74-77) recognising the many different names for bilby. We have since extended this text to help the reader understand that we use Ninu to mean the wild samples in addition to samples from the metapopulation where the bloodlines are mixed and sources unknown (see lines 98-104).

Line 74-77: *“The many First Nations across Australia have different names for bilby (Extended Data Table 1), but here we use Ninu to represent the Greater bilby as this is the name used by the Kiwirrkurra community (where most of our wild samples are from), and Yallara for the lesser bilby.”*

Line 98-104: *“The zoo-based populations were managed as two separate evolutionary units (NT/WA and QLD), until 2016 when they were combined into one metapopulation⁷, **resulting in the mixing of different bilby bloodlines from many Indigenous communities making it difficult to attribute metapopulation individuals to any particular Traditional Owner group.** Since 1996 Ninu from the zoo population have been released to large, fenced sanctuaries and islands, **noting that the concept of translocations is culturally sensitive to many Indigenous communities.**”*

For clarity line 166 now reads: *“Twelve Ninu from the metapopulation were used for the whole genome resequencing (WGR).....”*

- Line 95. The concept of translocation for many Aboriginal groups can be culturally sensitive – a statement to this effect could be added to this manuscript.

Response: Added text at line 104: “.....and islands, noting that the concept of translocations is culturally sensitive to many Indigenous communities.”

- Lines 147-153, it is likely that Indigenous Knowledge would support this concept of ‘boom and bust’. If possible and ICIP is supported an appropriate ‘Ninu’ story could be shared. This would demonstrate the two Knowledge systems side by side.

Response: Advice from our Indigenous liaison officer is that reproductive information is typically “women’s business” and so is not appropriate to share here. We will work towards the two Knowledge systems in our follow-up scat project to ascertain if we are able to obtain ICIP to share the information.

- Line 76. Consider replacing the term ‘ornaments’ with ‘cultural artefact or ‘used in cultural practice’.

Response: Changed text (line 79) to read “in cultural practices”

- Line 270 and Line 273 both community rangers and Indigenous rangers are used throughout; I suggest consistency using Indigenous rangers.

Response: Changed to read “Indigenous rangers” throughout the manuscript

- Lines 498-500, depending on how the above feedback is incorporated, this sentence will need revision.

Response: Text has been adjusted to read (line 404): “.....allowed us to sequence genomes for both the extant Greater bilby (called Ninu here) and the extinct.....”

In addition, line 408-411 has been changed to read: “Here we showcase the bilby’s unusual biology, in addition to their cultural value and importance. We recognise the many First Nations names for the species (Extended Data Table 1), and use the name Ninu here in recognition of the wild samples provided by the Kiwirrikurra Indigenous rangers.”

Decision Letter, first revision:

2nd April 2024

23Dear Carolyn,

I hope you're well. Vera is on leave at the moment, so I'm just sending this email on her behalf.

Thank you for submitting your revised manuscript "Extant and extinct bilby genomes combined with Indigenous knowledge improve conservation of a unique Australian marsupial" (NATECOLEVOL-23122946A). It has now been seen again by the original reviewers and their comments are below. The reviewers find that the paper has improved in revision, and therefore we'll be happy in principle to publish it in Nature Ecology & Evolution, pending minor revisions to satisfy the reviewers' final requests and to comply with our editorial and formatting guidelines.

We are now performing detailed checks on your paper and will send you a checklist detailing our editorial and formatting requirements within two weeks. Please do not upload the final materials and make any revisions until you receive this additional information from us.

[REDACTED]

Reviewer #1 (Remarks to the Author):

My concerns have been adequately addressed, and it is good to see more emphasis on collaboration with Indigenous rangers. Adding in the next steps for the scat protocol will be useful to future conservation researchers.

I recommend accept, but further submit the following quite minor changes as recommendations:

Pg4 Line 148:

"The Genome Landscape" to "The Genomic Landscape"

For the following change, I would further recommend linking to the management recommendations as you would reference an extended figure or table. "These recommendations included maximising genetic diversity and value of the metapopulation, sourcing wild bilbies, biobanking genetic samples and using the scat method to undertake a nationwide survey."

Reviewer #2 (Remarks to the Author):

The authors have done a very good job in carefully and thoroughly addressing the concerns raised in the reviews and revising the manuscript accordingly. I think the manuscript is now much improved,

24and conveys its relevant and interesting findings more clearly.

I only noticed a small typo in one of the revised sentences. So this can be fixed in the final version:

L.140: 'This comprehensive study arose for the need...' should likely read 'This comprehensive study arose from the need...'

Overall, I commend the authors on their efforts, and wish them luck in the continuation of this research and conservation program.

Reviewer #3 (Remarks to the Author):

Suitable for publication with NEE - well done.

Thank you for taking the time to address my comments and I appreciate challenge of balance with other reviewers suggestion not focusing on some of the Indigenous perspectives.

Our ref: NATECOLEVOL-23122946A

10th April 2024

Dear Dr. Hogg,

Thank you for your patience as we've prepared the guidelines for final submission of your Nature Ecology & Evolution manuscript, "Extant and extinct bilby genomes combined with Indigenous knowledge improve conservation of a unique Australian marsupial" (NATECOLEVOL-23122946A). Please carefully follow the step-by-step instructions provided in the attached file, and add a response in each row of the table to indicate the changes that you have made. Please also check and comment on any additional marked-up edits we have proposed within the text. Ensuring that each point is addressed will help to ensure that your revised manuscript can be swiftly handed over to our production team.

****We would like to start working on your revised paper, with all of the requested files and forms, as soon as possible (preferably within two weeks). Please get in contact with us immediately if you anticipate it taking more than two weeks to submit these revised files.****

If you have not done so already, please alert us to any related manuscripts from your group that are

25under consideration or in press at other journals, or are being written up for submission to other journals (see: <https://www.nature.com/nature-research/editorial-policies/plagiarism#policy-on-duplicate-publication> for details).

In recognition of the time and expertise our reviewers provide to Nature Ecology & Evolution's editorial process, we would like to formally acknowledge their contribution to the external peer review of your manuscript entitled "Extant and extinct bilby genomes combined with Indigenous knowledge improve conservation of a unique Australian marsupial". For those reviewers who give their assent, we will be publishing their names alongside the published article.

Nature Ecology & Evolution offers a Transparent Peer Review option for new original research manuscripts submitted after December 1st, 2019. As part of this initiative, we encourage our authors to support increased transparency into the peer review process by agreeing to have the reviewer comments, author rebuttal letters, and editorial decision letters published as a Supplementary item. When you submit your final files please clearly state in your cover letter whether or not you would like to participate in this initiative. Please note that failure to state your preference will result in delays in accepting your manuscript for publication.

Cover suggestions

We welcome submissions of artwork for consideration for our cover. For more information, please see our guide for cover artwork.

Nature Ecology & Evolution has now transitioned to a unified Rights Collection system which will allow our Author Services team to quickly and easily collect the rights and permissions required to publish your work. Approximately 10 days after your paper is formally accepted, you will receive an email in providing you with a link to complete the grant of rights. If your paper is eligible for Open Access, our Author Services team will also be in touch regarding any additional information that may be required to arrange payment for your article.

Please note that *Nature Ecology & Evolution* is a Transformative Journal (TJ). Authors may publish their research with us through the traditional subscription access route or make their paper immediately open access through payment of an article-processing charge (APC). Authors will not be required to make a final decision about access to their article until it has been accepted. Find out more about Transformative Journals

Authors may need to take specific actions to achieve compliance with funder and institutional open access mandates. If your research is supported by a funder that requires

26immediate open access (e.g. according to Plan S principles) then you should select the gold OA route, and we will direct you to the compliant route where possible. For authors selecting the subscription publication route, the journal's standard licensing terms will need to be accepted, including <https://www.nature.com/nature-portfolio/editorial-policies/self-archiving-and-license-to-publish>. Those licensing terms will supersede any other terms that the author or any third party may assert apply to any version of the manuscript.

Please use the following link for uploading these materials:
[REDACTED]

[REDACTED]

Reviewer #1:

Remarks to the Author:

My concerns have been adequately addressed, and it is good to see more emphasis on collaboration with Indigenous rangers. Adding in the next steps for the scat protocol will be useful to future conservation researchers.

I recommend accept, but further submit the following quite minor changes as recommendations:

Pg4 Line 148:

"The Genome Landscape" to "The Genomic Landscape"

For the following change, I would further recommend linking to the management recommendations as you would reference an extended figure or table. "These recommendations included maximising genetic diversity and value of the metapopulation, sourcing wild bilbies, biobanking genetic samples and using the scat method to undertake a nationwide survey."

Reviewer #2:

Remarks to the Author:

The authors have done a very good job in carefully and thoroughly addressing the concerns raised in the reviews and revising the manuscript accordingly. I think the manuscript is now much improved, and conveys its relevant and interesting findings more clearly.

27I only noticed a small typo in one of the revised sentences. So this can be fixed in the final version:

L.140: 'This comprehensive study arose for the need...' should likely read 'This comprehensive study arose from the need...'

Overall, I commend the authors on their efforts, and wish them luck in the continuation of this research and conservation program.

Reviewer #3:

Remarks to the Author:

Suitable for publication with NEE - well done.

Thank you for taking the time to address my comments and I appreciate challenge of balance with other reviewers suggestion not focusing on some of the Indigenous perspectives.

Author Rebuttal, first revision:

Response to Reviewers – Second round

Reviewer #1:

I recommend accept, but further submit the following quite minor changes as recommendations:

Pg4 Line 148:

"The Genome Landscape" to "The Genomic Landscape"

Response: Changed to "The genomic landscape"

For the following change, I would further recommend linking to the management recommendations as you would reference an extended figure or table. "These recommendations included maximising genetic diversity and value of the metapopulation, sourcing wild bilbies, biobanking genetic samples and using the scat method to undertake a nationwide survey."

Response: Added (Supplementary Note 2.6) at the end of the sentence.

Reviewer #2:

I only noticed a small typo in one of the revised sentences. So this can be fixed in the final version:

L.140: 'This comprehensive study arose for the need...' should likely read 'This comprehensive study arose from the need...'

Response: Changed the word “for” to “from” so the sentence now reads “This comprehensive study arose from the need to understand.....”

Reviewer #3:

No comments needing action

Decision Letter, second revision:

3rd May 2024

Dear Carolyn,

We are pleased to inform you that your Article entitled "Extant and extinct bilby genomes combined with Indigenous knowledge improve conservation of a unique Australian marsupial", has now been accepted for publication in Nature Ecology & Evolution.

Over the next few weeks, your paper will be copyedited to ensure that it conforms to Nature Ecology and Evolution style. Once your paper is typeset, you will receive an email with a link to choose the appropriate publishing options for your paper and our Author Services team will be in touch regarding any additional information that may be required

Due to the importance of these deadlines, we ask you please us know now whether you will be difficult to contact over the next month. If this is the case, we ask you provide us with the contact information (email, phone and fax) of someone who will be able to check the proofs on your behalf, and who will be available to address any last-minute problems . Once your paper has been scheduled for online publication, the Nature press office will be in touch to confirm the details.

Acceptance of your manuscript is conditional on all authors' agreement with our publication policies (see www.nature.com/authors/policies/index.html). In particular your manuscript must not be published elsewhere and there must be no announcement of the work to any media outlet until the

29publication date (the day on which it is uploaded onto our web site).

Please note that *Nature Ecology & Evolution* is a Transformative Journal (TJ). Authors may publish their research with us through the traditional subscription access route or make their paper immediately open access through payment of an article-processing charge (APC). Authors will not be required to make a final decision about access to their article until it has been accepted. Find out more about Transformative Journals

Authors may need to take specific actions to achieve compliance with funder and institutional open access mandates. If your research is supported by a funder that requires immediate open access (e.g. according to Plan S principles) then you should select the gold OA route, and we will direct you to the compliant route where possible. For authors selecting the subscription publication route, the journal's standard licensing terms will need to be accepted, including [a href="https://www.nature.com/nature-portfolio/editorial-policies/self-archiving-and-license-to-publish"](https://www.nature.com/nature-portfolio/editorial-policies/self-archiving-and-license-to-publish). Those licensing terms will supersede any other terms that the author or any third party may assert apply to any version of the manuscript.

We welcome the submission of potential cover material (including a short caption of around 40 words) related to your manuscript; suggestions should be sent to Nature Ecology & Evolution as electronic files (the image should be 300 dpi at 210 x 297 mm in either TIFF or JPEG format). Please note that such pictures should be selected more for their aesthetic appeal than for their scientific content, and that colour images work better than black and white or grayscale images. Please do not try to design a cover with the Nature Ecology & Evolution logo etc., and please do not submit composites of images related to your work. I am sure you will understand that we cannot make any promise as to whether any of your suggestions might be selected for the cover of the journal.

30You can generate the link yourself when you receive your article DOI by entering it here: <http://authors.springernature.com/share>.

[REDACTED]

P.S. Click on the following link if you would like to recommend Nature Ecology & Evolution to your librarian <http://www.nature.com/subscriptions/recommend.html#forms>

** Visit the Springer Nature Editorial and Publishing website at www.springernature.com/editorial-and-publishing-jobs for more information about our career opportunities. If you have any questions please click here.**

Final Decision Letter:

Dear Carolyn,

We are pleased to inform you that your Article entitled "Extant and extinct bilby genomes combined with Indigenous knowledge improve conservation of a unique Australian marsupial", has now been accepted for publication in Nature Ecology & Evolution.

Over the next few weeks, your paper will be copyedited to ensure that it conforms to Nature Ecology and Evolution style. Once your paper is typeset, you will receive an email with a link to choose the appropriate publishing options for your paper and our Author Services team will be in touch regarding any additional information that may be required

Due to the importance of these deadlines, we ask you please us know now whether you will be difficult to contact over the next month. If this is the case, we ask you provide us with the contact information (email, phone and fax) of someone who will be able to check the proofs on your behalf, and who will be available to address any last-minute problems . Once your paper has been scheduled for online publication, the Nature press office will be in touch to confirm the details.

Acceptance of your manuscript is conditional on all authors' agreement with our publication policies (see www.nature.com/authors/policies/index.html). In particular your manuscript must not be published elsewhere and there must be no announcement of the work to any media outlet until the publication date (the day on which it is uploaded onto our web site).

31Please note that *Nature Ecology & Evolution* is a Transformative Journal (TJ). Authors may publish their research with us through the traditional subscription access route or make their paper immediately open access through payment of an article-processing charge (APC). Authors will not be required to make a final decision about access to their article until it has been accepted. Find out more about Transformative Journals

Authors may need to take specific actions to achieve compliance with funder and institutional open access mandates. If your research is supported by a funder that requires immediate open access (e.g. according to Plan S principles) then you should select the gold OA route, and we will direct you to the compliant route where possible. For authors selecting the subscription publication route, the journal's standard licensing terms will need to be accepted, including [a href="https://www.nature.com/nature-portfolio/editorial-policies/self-archiving-and-license-to-publish"](https://www.nature.com/nature-portfolio/editorial-policies/self-archiving-and-license-to-publish). Those licensing terms will supersede any other terms that the author or any third party may assert apply to any version of the manuscript.

We welcome the submission of potential cover material (including a short caption of around 40 words) related to your manuscript; suggestions should be sent to Nature Ecology & Evolution as electronic files (the image should be 300 dpi at 210 x 297 mm in either TIFF or JPEG format). Please note that such pictures should be selected more for their aesthetic appeal than for their scientific content, and that colour images work better than black and white or grayscale images. Please do not try to design a cover with the Nature Ecology & Evolution logo etc., and please do not submit composites of images related to your work. I am sure you will understand that we cannot make any promise as to whether any of your suggestions might be selected for the cover of the journal.

You can generate the link yourself when you receive your article DOI by entering it here: <http://authors.springernature.com/share>.

[REDACTED]

P.S. Click on the following link if you would like to recommend Nature Ecology & Evolution to your librarian <http://www.nature.com/subscriptions/recommend.html#forms>

** Visit the Springer Nature Editorial and Publishing website at www.springernature.com/editorial-and-publishing-jobs for more information about our career opportunities. If you have any questions please click here.**